# Contextual Linear Optimization with Bandit Feedback

**Yichun Hu**[1*]    **Nathan Kallus**[1*]    **Xiaojie Mao**[2*]    **Yanchen Wu**[2*]

[1] Cornell University    [2] Tsinghua University

{yh767, kallus}@cornell.edu
maoxj@sem.tsinghua.edu.cn
wu-yc23@mails.tsinghua.edu.cn

## Abstract

Contextual linear optimization (CLO) uses predictive contextual features to reduce uncertainty in random cost coefficients and thereby improve average-cost performance. An example is the stochastic shortest path problem with random edge costs (e.g., traffic) and contextual features (e.g., lagged traffic, weather). Existing work on CLO assumes the data has fully observed cost coefficient vectors, but in many applications, we can only see the realized cost of a historical decision, that is, just one projection of the random cost coefficient vector, to which we refer as bandit feedback. We study a class of offline learning algorithms for CLO with bandit feedback, which we term induced empirical risk minimization (IERM), where we fit a predictive model to directly optimize the downstream performance of the policy it induces. We show a fast-rate regret bound for IERM that allows for misspecified model classes and flexible choices of the optimization estimate, and we develop computationally tractable surrogate losses. A byproduct of our theory of independent interest is fast-rate regret bound for IERM with full feedback and misspecified policy class. We compare the performance of different modeling choices numerically using a stochastic shortest path example and provide practical insights from the empirical results.

## 1  Introduction

Contextual linear optimization (CLO) models the use of predictive features (context variables) to improve decision making in linear optimization with random coefficients. In CLO, we consider a decision $z \in \mathcal{Z}$ that incurs an uncertain cost $Y^\top z$ determined by a random cost vector $Y \in \mathbb{R}^d$ that is not observed at decision time. We do, however, observe predictive features $X \in \mathbb{R}^p$ prior to decision, which help reduce uncertainty. CLO can be expressed either as a contextual stochastic optimization problem or a linear optimization problem where the cost vector is a conditional expectation:

$$v^*(x) = \min_{z \in \mathcal{Z}} \mathbb{E}\big[Y^\top z \mid X = x\big] = \min_{z \in \mathcal{Z}} f_0(x)^\top z, \quad \text{where} \quad f_0(x) = \mathbb{E}[Y \mid X = x]. \quad (1)$$

We assume throughout that $\mathcal{Z}$ is a polytope ($\mathcal{Z} = \text{Conv}(\mathcal{Z}^\angle)$ for some vertex set $|\mathcal{Z}^\angle| < \infty$ with $\sup_{z \in \mathcal{Z}^\angle} \|z\| \leq B$) and $Y$ is bounded (without loss of generality, $Y \in \mathcal{Y} = \{y : \|y\| \leq 1\}$).

CLO has been the focus of much recent work [e.g., 9, 11, 14, 27, 30], due to its relevance to data-driven operational decision making in many applications such as network optimization, portfolio optimization, product recommendation, etc. The key question in these is how to use *data* to learn a good policy, $\pi : \mathbb{R}^p \to \mathcal{Z}$, mapping feature observations to effective decisions. All of this literature studies an offline setting, where a batch of existing data $(X_i, Y_i)$ for $i = 1, \ldots, n$ is observed. These data are viewed as random draws from the joint distribution of $(X, Y)$, and the goal is to learn an effective decision policy from them. Crucially, this assumes that we fully observe the random cost

---

*Authors are listed in alphabetical order. Correspondence to Xiaojie Mao: maoxj@sem.tsingua.edu.cn.

38th Conference on Neural Information Processing Systems (NeurIPS 2024).

vector $Y_i$'s, meaning that we fully know the corresponding costs $Y_i^\top z$ for *any* potential decision $z$. This may be unrealistic in many applications.

Consider for example the instantiation of Eq. (1) as a stochastic shortest path problem of transporting one unit from a start node to a desired end node, where $Y$ contains the travel time on each of $d$ edges, $z$ the binary decision whether to traverse each edge, and $\mathcal{Z}$ constrains flow conservation. The data we would need in order to apply existing approaches here would consist of simultaneous observations of the travel times on every single edge, encapsulated in the elements of $Y_i$'s. However, such ideal observations may not be available. Indeed, modern travel time prediction models are often based on data from historical individual trips. For example, such is the case with Uber Movement data [26, 33], which is based on the total length of historical rides. Namely, instead of observing the entire $Y_i$, we *only* observe the total travelling time $C_i = Y_i^\top Z_i$ for the path $Z_i$ in a historical trip $i$. We term this observation model *bandit feedback* as it corresponds to observing the cost only of a given decision and not the counterfactual cost of any alternative decision, as in bandit problems [25].

In this paper, we study the contextual linear optimization problem with bandit feedback and make several contributions. First, we adapt the end-to-end induced empirical risk minimization approach to the bandit feedback setting. This approach allows us to optimize decision policies by directly targeting the decision objective. We provide three different methods to identify the expected cost of any given policy and show how to estimate the expected decision cost from bandit-feedback data. Second, we derive upper bounds on the regret (*i.e.*, decision sub-optimality) of the policy that minimizes the estimated objective. Our regret analysis accounts for the misspecification of the policy class and incorporates a margin condition that potentially enables a faster regret rate. This significantly extends existing theory for full-feedback CLO, and as a byproduct, we provide a novel fast rate bound for full-feedback CLO with misspecification. Finally, we demonstrate that existing surrogate losses for full-feedback CLO (such as the well-known SPO+ loss in [11]) can be adapted to our setting, enabling efficient policy optimization. We empirically test this in simulated shortest path problems and provide practical insights.

## 1.1 Background: The Full Feedback Setting

We first review two major approaches to the CLO problem with full feedback: an estimate-then-optimize (ETO) approach and an end-to-end induced empirical risk minimization (IERM) approach. To this end, we need to define the generic plug-in policy $\pi_f$ for any given $f : \mathbb{R}^p \to \mathbb{R}^d$:

$$\pi_f(x) \in \arg\min_{z \in \mathcal{Z}} f(x)^\top z. \tag{2}$$

Note that for any given covariate value $x$, this corresponds to a linear programming problem with coefficients given by $f(x)$. An optimal policy in Eq. (1) corresponds to $\pi_{f_0}$. Without loss of generality, we restrict the value of $\pi_f(x)$ to the set of vertices $\mathcal{Z}^\angle$ of the polytope $\mathcal{Z}$ with ties broken according to some fixed rule (e.g., a total ordering over $\mathcal{Z}^\angle$ such as lexicographic).

The ETO approach starts with estimating $f_0$ in Eq. (1) by any supervised learning method for predicting $Y$ given $X$. This can, *e.g.*, be implemented by minimizing a prediction fitness criterion (e.g., sum of squared errors) over a hypothesis class of functions $\mathcal{F} \subseteq [\mathbb{R}^p \to \mathbb{R}^d]$ (e.g., linear functions, decision trees, neural networks). After training an estimator $\hat{f}$ of $f_0$, the ETO approach then makes decisions according to the induced policy $\pi_{\hat{f}}$. This approach is a straightforward two-step approach, but it ignores the downstream optimization task in choosing $\hat{f} \in \mathcal{F}$.

In contrast, some recent literature propose to integrate the estimation and the optimization, directly searching for a decision policy to minimize the decision cost. Following [14], we consider an IERM formulation that minimizes the sample average cost over the class of plug-in policies induced by $\mathcal{F}$:

$$\hat{\pi}^{\mathrm{F}} \in \arg\min_{\pi \in \Pi_\mathcal{F}} \tfrac{1}{n} \sum_{i=1}^n Y_i^\top \pi(X_i), \quad \text{where } \Pi_\mathcal{F} = \{\pi_f : f \in \mathcal{F}\}. \tag{3}$$

This approach is end-to-end in that it directly targets the decision-making objective. Recent literature demonstrates that end-to-end approaches like IERM often outperform the ETO approach [12]. The benefits of end-to-end approaches are significant especially in the misspecified setting - that is, when the function class $\mathcal{F}$ fails to contain the true expected cost function $f_0$, and the policy class $\Pi_\mathcal{F}$ does not include the globally optimal decision policy $\pi_{f_0}$ [e.g., 11, 14]. While Eq. (3) involves a challenging bi-level optimization due to the definition of $\Pi_\mathcal{F}$, practical computational approximations have been proposed [11, 16].

Nonetheless, the IERM formulation in Eq. (3) requires observing the full feedback $Y_i$'s in the data. In this paper, we study how to extend this approach to the setting with bandit feedback.

## 1.2 Our Problem: CLO with Bandit Feedback

We now formally define the data generating process in the bandit feedback setting. Assume we have an offline dataset consisting of $n$ data points $\mathcal{D} = \{(X_1, Z_1, C_1), \ldots, (X_n, Z_n, C_n)\}$ that are independent draws from a distribution $P$ on $(X, Z, C)$ generated in the following way. We first draw $(X, Z, Y)$ where the $(X, Y)$ distribution is as in the full feedback setting and $Z$ is generated according to some historical decision policies for data collection. Then we set $C = Z^\top Y$ and omit the full $Y$. Below we impose some basic assumptions on the generating process of the historical decision $Z$.

**Assumption 1.** *The data generating process satisfies the following two properties:*

1. *(Ignorability)* $\mathbb{E}[C \mid Z, X] = Z^\top f_0(X)$ *(which could follow from $Z \perp\!\!\!\perp Y \mid X$).*

2. *(Coverage)* $\inf_{z \in \mathrm{span}(\mathcal{Z}): \|z\|=1} \mathbb{E}[(Z^\top z)^2 \mid X] > 0$ *almost surely.*

The ignorability condition is a common assumption that plays a fundamental role in learning with partial feedback and causal inference (see the literature in Section 1.3). This condition requires that the assignments of the historical decisions $Z$ do not depend on any unobserved factor potentially dependent on $Y$. The coverage condition requires that given the covariates $X$, historical decisions $Z$ can explore all directions of the linear span of the constraint set $\mathcal{Z}$; otherwise it may be impossible to evaluate certain policies from the observed costs of historical decisions. This is an analogue of the overlap assumption in learning with partial feedback and causal inference [17].

The learning task is to use these data $\mathcal{D}$ to come up with a data-driven policy $\hat{\pi}(x)$ that has low regret

$$\mathrm{Reg}(\pi) = \mathbb{E}_X\big[f_0(X)^\top \pi(X) - \min_{z \in \mathcal{Z}} f_0(X)^\top z\big] = V(\pi) - V(\pi_{f_0}), \qquad (4)$$

where $V(\pi) = \mathbb{E}_X\big[f_0(X)^\top \pi(X)\big]$.

## 1.3 Background: Off-Policy Evaluation and Optimization

An important special case is when $\mathcal{Z}^\angle = \{(1, 0, \ldots, 0), \ldots, (0, \ldots, 0, 1)\}$ is the canonical basis and $\mathcal{Z}$ is the simplex. This corresponds to choosing one of $d$ actions (or randomizing between them). The full-feedback problem in this case is cost-sensitive classification [10]. In the bandit feedback case, with the restriction $Z \in \mathcal{Z}^\angle$, it is known as a logged or offline contextual bandit, and a long line of literature studies the problems of estimating and optimizing $V(\pi)$ [1, 8, 18–20, 23, 24, 32, 34–36, 42–45]. It is closely related to causal inference, viewing each component of $Y$ as the potential outcome for the corresponding action (or treatment), where we only see the realized outcome $C$ corresponding to the factual action with index 1 in $Z \in \mathcal{Z}^\angle$ and see nothing about counterfactual actions $\mathcal{Z}^\angle \backslash \{Z\}$. Two key differentiating characteristics of our problem with a general polytope constraint $\mathcal{Z}$ are the opportunity to leverage the linear cost structure to extrapolate from the costs of historical decisions to costs of other decisions, and the presence of non-trivial constraints on the output of decision policies.

Going beyond just finite discrete arms, [21, 22, 29] consider the logged contextual bandit with a *continuum* of arms, which is not generally representable in terms of the problem in Section 1.2. These works make no restrictions on the relationship between the expected potential outcome and the corresponding action except generic smoothness, and leverage nonparametric approaches such as kernel weighting. Closest to our work is [6], which imposes a *semiparametric* assumption on this relationship. This includes our problem (Section 1.2) as a special case under the restriction that, conditional on $X$, expected outcomes are linear in actions (note that our $\mathbb{E}[C \mid Z, X]$ is linear in $Z$ according to ignorability in Section 1.2). Relative to this work, our unique contributions are the use of induced policy classes to naturally deal with the constraints in $\mathcal{Z}$, the adaptation of computational approximation such as SPO+, and obtaining the fast rates for the regret under misspecification.

## 2 Induced Empirical Risk Minimization with Bandit Feedback

Our IERM approach in the bandit setting follows the same idea of Eq. (3), directly minimizing an estimate of expected costs. However, since we do not observe $Y_i$ in our data, it is not as straightforward as a sample average, $\sum_{i=1}^n Y_i^\top \pi(X_i)/n$. Instead, we need alternative ways to identify $V(\pi)$.

Define $\Sigma_0(X) = \mathbb{E}\left[ZZ^\top \mid X\right]$ and recall $f_0(X) = \mathbb{E}[Y \mid X]$, which we will use to characterize the policy value $V(\pi)$. We refer to these functions as *nuisance functions* following the existing literature on off-policy evaluation and learning in contextual bandits or reinforcement learning [39], because they are not directly used for decision-making but serve merely as intermediaries for evaluating the policy value. Let $\theta(x, z, c; f, \Sigma)$ be a score function such that

$$\mathbb{E}_P\left[\theta(X, Z, C; f_0, \Sigma_0)^\top \pi(X)\right] = V(\pi), \quad \forall \text{ fixed policy } \pi, \tag{5}$$

where $\mathbb{E}_P$ denotes taking expectation over $P$. Proposition 1 summarizes a few possible choices.

**Proposition 1.** *The following choices of $\theta(x, z, c; f, \Sigma)$ all satisfy Eq. (5):*

1. *(Direct Method) $\theta_{DM}(x, z, c; f, \Sigma) = f(x)$;*

2. *(Inverse Spectral Weighting) $\theta_{ISW}(x, z, c; f, \Sigma) = \Sigma^\dagger(x)zc$;*

3. *(Doubly Robust) $\theta_{DR}(x, z, c; f, \Sigma) = f(x) + \Sigma^\dagger(x)z(c - z^\top f(x))$.*

*Here $\Sigma^\dagger$ denotes the Moore–Penrose pseudo-inverse of matrix $\Sigma$.*

The doubly robust identification in Proposition 1 is a generalization of the identification in [6] in that we allow for rank-deficient $\Sigma_0$ by using a pseudo-inverse. Otherwise, our work significantly differs from [6] in terms of policy class specification, computation and theoretical analyses (see Section 1.3). Note that identification in Proposition 1 exploits the linearity of the decision cost, which significantly improve upon algorithms that ignore the linear structure and naively extend existing offline bandit learning using discrete actions $\mathcal{Z}^\perp$; see Section 5 and Appendix C.2 for numerical evidence.

It remains to estimate Eq. (5), since we know neither $P$ nor the nuisance functions $f_0, \Sigma_0$. Following the approach of [1, 45] for logged contextual bandits, we adapt the cross-fitting procedure advocated in [5]. For simplicity, we focus on the twofold version and assume $n$ is even. The extension to $K$-fold version is straightforward.

First, we split $\mathcal{D}$ into two equal-sized subsamples, $\mathcal{D}_1$ and $\mathcal{D}_2$, each with $n/2$ data points. We can use $\mathcal{D}_1$ as an auxiliary sample to estimate the nuisance functions $f_0$ and $\Sigma_0$, denoting the estimates as $\hat{f}_1$ and $\hat{\Sigma}_1$. We then use $\mathcal{D}_2$ as a main sample to obtain an estimate of $V(\pi)$ using $\hat{f}_1$ and $\hat{\Sigma}_1$:

$$\tfrac{2}{n} \sum_{i \in \mathcal{D}_2} \theta(X_i, Z_i, C_i; \hat{f}_1, \hat{\Sigma}_1)^\top \pi(X_i).$$

Of course, we can switch the roles of $\mathcal{D}_1$ and $\mathcal{D}_2$, *i.e.*, we use $\mathcal{D}_2$ to get nuisance estimates $\hat{f}_2$ and $\hat{\Sigma}_2$, then use $\mathcal{D}_1$ to estimate $V(\pi)$. Finally, given an induced policy class $\Pi_\mathcal{F}$, the IERM policy $\hat{\pi}$ minimizes the average of the above two $V(\pi)$ estimates over $\Pi_\mathcal{F}$:

$$\hat{\pi} \in \arg\min_{\pi \in \Pi_\mathcal{F}} \tfrac{1}{n} \sum_{j=1}^{2} \sum_{i \in \mathcal{D}_j} \theta(X_i, Z_i, C_i; \hat{f}_{3-j}, \hat{\Sigma}_{3-j})^\top \pi(X_i). \tag{6}$$

**Remark 1** (Estimation of $f_0$). *The estimator $\hat{f}(x)$ can be obtained by minimizing the least square loss over some appropriate function class $\mathcal{F}^N$:*

$$\hat{f} \in \arg\min_{f \in \mathcal{F}^N} \sum_{i \in \mathcal{D}} \left(C_i - Z_i^\top f(X_i)\right)^2. \tag{7}$$

*Note that two places in Eq. (6) require the specification of a function class for modeling $f_0$: the $\mathcal{F}$ class used to induce the policy class $\Pi_\mathcal{F}$, and the $\mathcal{F}^N$ class used to construct nuisance estimators to evaluate the expected cost of induced policies. In practice, we do not need to use the same class for $\mathcal{F}$ and $\mathcal{F}^N$. In fact, it might be desirable to use a more flexible function class for $\mathcal{F}^N$ to make sure it is well-specified for accurate policy evaluation, and a simpler class for $\mathcal{F}$ to make the policy optimization more tractable. We numerically test out different choices of $\mathcal{F}$ and $\mathcal{F}^N$ in Section 5.*

**Remark 2** (Estimation of $\Sigma_0$). *There are multiple ways to estimate $\Sigma_0$. For example, [6] suggest estimating $\Sigma_0$ by running a multi-task regression for all $(j, k)$ entries to the matrix over some appropriate hypothesis spaces $\mathcal{S}_{jk}$: $\hat{\Sigma} = \arg\min_{\Sigma_{11} \in \mathcal{S}_{11}, \ldots, \Sigma_{dd} \in \mathcal{S}_{dd}} \sum_{i \in \mathcal{D}} \|Z_i Z_i^\top - \Sigma(X_i)\|_{Fro}^2$. To ensure a positive semi-definite estimator, we may posit each hypothesis $\Sigma$ to be the outer product of some matrix-valued hypothesis of appropriate dimension. Alternatively, we may only need to consider finitely many feasible decisions $z_1, \ldots, z_m$, such as the feasible paths for stochastic shortest path problems (see experiments in Section 5) or more generally the vertices in $\mathcal{Z}^\perp$. Then we can first estimate the propensity scores $e_0(z \mid X)$ for $z = z_1, \ldots, z_m$ using an suitable estimator $\hat{e}(z \mid X)$ and then estimate $\Sigma_0(X) = \sum_{j=1}^{m} z_j z_j^\top e_0(z_j \mid X)$ by $\hat{\Sigma}(X) = \sum_{j=1}^{m} z_j z_j^\top \hat{e}(z_j \mid X)$.*

# 3 Theoretical Analysis

In this section, we provide a theoretical regret analysis for the IERM approach with bandit feedback, allowing for model misspecification in the induced policy class $\Pi_{\mathcal{F}}$. That is, we allow the globally optimal policy $\pi_{f_0}$ to be not included in the class $\Pi_{\mathcal{F}}$. We derive an upper bound on the regret of the IERM approach, in terms of the complexity of the policy class $\Pi_{\mathcal{F}}$, its misspecification bias, and the estimation errors of nuisance functions.

Before stating the main theorem, we define a few important notations. Let $\tilde{\pi}^*$ be the best-in-class policy that minimizes expected regret over $\Pi_{\mathcal{F}}$:

$$\tilde{\pi}^* \in \arg\min_{\pi \in \Pi_{\mathcal{F}}} \mathbb{E}\big[f_0(X)^\top \pi(X)\big].$$

We note that $\tilde{\pi}^*$ can be different from the global optimal policy $\pi_{f_0}$ if $\pi_{f_0} \notin \Pi_{\mathcal{F}}$, and $\mathrm{Reg}(\tilde{\pi}^*)$ characterizes the extent of misspecification in the induced policy class $\Pi_{\mathcal{F}}$. For $j = 1, 2$, we define the function class

$$\mathcal{G}_j = \left\{ (x, z, c) \to \frac{\theta(x, z, c; \hat{f}_j, \hat{\Sigma}_j)^\top (\pi(x) - \tilde{\pi}^*(x))\rho}{2B\Theta} : \pi \in \Pi_{\mathcal{F}}, \rho \in [0, 1] \right\}. \tag{8}$$

For any function class $\mathcal{G} : \mathcal{X} \times \mathcal{Z} \times \mathcal{C} \to \mathbb{R}$, we define the local Rademacher complexity as

$$\mathcal{R}_n(\mathcal{G}, r) = \mathbb{E}\Big[\sup_{g \in \mathcal{G}, \|g\|_2 \leq r} \big| \tfrac{1}{n} \sum_{i=1}^n \epsilon_i g(X_i, Z_i, C_i)\big|\Big],$$

where $\epsilon_i$ are i.i.d. Rademacher variables, and $\|g\|_2 = \sqrt{\mathbb{E}_P[g^2(X, Z, C)]}$.

Throughout Section 3, we impose two assumptions. Assumption 2 concerns the algorithms we use to estimate the nuisance functions $f_0, \Sigma_0$. It requires that the nuisance estimates are close enough to their true values, and $\|\theta(x, z, c; \hat{f}, \hat{\Sigma})\|$ is bounded.

**Assumption 2** (Nuisance Estimation). *The nuisance estimators $\hat{f}, \hat{\Sigma}$ trained on a sample of size $n$ satisfy that $\|\theta(x, z, c; \hat{f}, \hat{\Sigma})\| \leq \Theta$ for all $x, z, c$, and for any $\delta \in (0, 1)$ and $\pi \in \Pi_{\mathcal{F}}$, with probability at least $1 - \delta$,*

$$\mathbb{E}_P\left[\Big(\theta(X, Z, C; f_0, \Sigma_0) - \theta(X, Z, C; \hat{f}, \hat{\Sigma})\Big)^\top (\pi(X) - \tilde{\pi}^*(X))\right] \leq \mathrm{Rate}^{\mathbf{N}}(n, \delta).$$

In Section 3.1, we will relate $\mathrm{Rate}^{\mathbf{N}}(n, \delta)$ to the errors in estimating the individual nuisance functions $f_0$ and $\Sigma_0$ for different choices of score function $\theta$.

Assumption 3, which we term the margin condition, controls the density of the sub-optimality gap near zero in the CLO problem instance and allows us to get faster regret rates. This type of condition was originally considered in the binary classification literature [2, 38]. It is recently extended to contextual linear optimization by [14], which we describe below.

**Assumption 3** (Margin Condition). *Let $\mathcal{Z}^*(x) = \arg\min_{z \in \mathcal{Z}} f_0(x)^\top z$, and $\Delta(x) = \inf_{z \in \mathcal{Z} \angle \setminus \mathcal{Z}^*(x)} f_0(x)^\top z - \inf_{z \in \mathcal{Z}} f_0(x)^\top z$ if $\mathcal{Z}^*(x) \neq \mathcal{Z}$ and $\Delta(x) = 0$ otherwise. Assume for some $\alpha, \gamma \geq 0$,*

$$\mathbb{P}_X(0 < \Delta(X) \leq \delta) \leq (\gamma\delta/B)^\alpha \quad \forall \delta > 0.$$

Lemmas 4 and 5 in [15] show that Assumption 3 generally holds with $\alpha = 1$ for sufficiently well-behaved $f_0$ and continuous $X$ and with $\alpha = \infty$ for discrete $X$. Moreover, any CLO instance trivially satisfies $\alpha = 0$. Overall, a larger $\alpha$ means that the sub-optimality gap between the best and second-best decisions tends to be large in more contexts, so it is easier to distinguish the optimal decisions from others. We will show that a larger $\alpha$ parameter could lead to faster regret rates.

## 3.1 Main Theorem

We now provide an upper bound on the regret of the IERM policy $\hat{\pi}$ in Eq. (6).

**Theorem 1.** *Suppose Assumptions 2 and 3 hold, and $\mathcal{Z}^*(X)$ defined in Assumption 3 is a singleton almost surely. Suppose there exists a positive number $\tilde{r}$ (that depends on $n$) that upper bounds the critical radii of the function classes $\mathcal{G}_1, \mathcal{G}_2$ almost surely (i.e., $\mathcal{R}_{n/2}(\mathcal{G}_1, \tilde{r}) \leq \tilde{r}^2$ and $\mathcal{R}_{n/2}(\mathcal{G}_2, \tilde{r}) \leq$*

$\tilde{r}^2$) and satisfies the inequalities $3n\tilde{r}^2/128 \geq \log\log_2(1/\tilde{r})$, and $2\exp\left(-3n\tilde{r}^2/128\right) \leq \delta/4$. Then, there exists a positive constant $\tilde{C}(\alpha, \gamma)$ such that with probability at least $1 - \delta$, we have

$$Reg(\hat{\pi}) \leq B\left(12\Theta\sqrt{\tilde{C}(\alpha,\gamma)}\tilde{r}\right)^{\frac{2\alpha+2}{\alpha+2}} + 24B\Theta\left(\sqrt{\tilde{C}(\alpha,\gamma)}\left(\frac{Reg(\tilde{\pi}^*)}{B}\right)^{\frac{\alpha}{2(1+\alpha)}}\tilde{r} + \tilde{r}^2\right)$$
$$+ 2Rate^{\textbf{N}}(n/2, \delta/4) + 2Reg(\tilde{\pi}^*).$$

The upper bound in Theorem 1 involves several different types of error terms. The first type is the critical radii $\tilde{r}$ that characterize the complexity of the function classes $\mathcal{G}_1, \mathcal{G}_2$. This is a common complexity measure for function classes in statistics and machine learning [41]. For example, as we will discuss below, the critical radii scale as $\tilde{O}(\sqrt{\eta/n})$ if $\mathcal{G}_1, \mathcal{G}_2$ are VC subgraph classes of dimension $\eta$. The second type is the term $Rate^{\textbf{N}}(n/2, \delta/4)$ resulted from the errors in estimating the nuisance functions $f_0, \Sigma_0$. Similar nuisance estimation errors also appear in the previous literature on offline contextual bandit learning [1, 5, 45] or more general learning problems that involve nuisance functions [13]. Below we will further discuss this term for different choices of score functions $\theta$. The third type is the misspecification error term $Reg(\tilde{\pi}^*)$. It is natural to expect a bigger decision regret when the misspecification error $Reg(\tilde{\pi}^*)$ is higher. In particular, when the policy class $\Pi_{\mathcal{F}}$ is well-specified, i.e., $\pi_{f_0} \in \Pi_{\mathcal{F}}$, we would have $Reg(\tilde{\pi}^*) = 0$, and the regret upper bound would scale with the function class complexity through a fast rate $O(\tilde{r}^{\frac{2\alpha+2}{\alpha+2}})$. For VC subgraph classes with $\tilde{r} = \tilde{O}(\sqrt{\eta/n})$, the rate can range from $\tilde{O}(\sqrt{\eta/n})$ for $\alpha = 0$ to $\tilde{O}(\eta/n)$ for $\alpha = \infty$. However, if the policy class is misspecified so that $Reg(\tilde{\pi}^*) > 0$, then the dominating term related to $\tilde{r}$ would be a slow rate $O(\tilde{r})$. This reveals an interesting phase transition between the correctly specified and misspecified settings, which is not discovered in the previous theory that considers only well-specification [14].

**Remark 3.** *The constant coefficients in the regret upper bound can be improved when $2(1 + \alpha)/\alpha$ is an integer (which accommodates the case of $\alpha = 1$ justified in [15]):*

$$Reg(\hat{\pi}) \leq B\left(12\Theta\sqrt{\tilde{C}(\alpha,\gamma)}\tilde{r}\right)^{\frac{2\alpha+2}{\alpha+2}} + \frac{24(\alpha+1)}{\alpha+2}B\Theta\left(\sqrt{\tilde{C}(\alpha,\gamma)}\left(\frac{Reg(\tilde{\pi}^*)}{B}\right)^{\frac{\alpha}{2(1+\alpha)}}\tilde{r} + \tilde{r}^2\right)$$
$$+ \frac{2\alpha+2}{\alpha+2}Rate^{\textbf{N}}(n/2, \delta/4) + Reg(\tilde{\pi}^*).$$

*Notably, the constant in front of the misspecification error $Reg(\tilde{\pi}^*)$ becomes 1 instead of 2, which we believe is tight. This upper bound follows from nearly the same proof as Theorem 1 except that it handles an inequality slightly differently. Specifically, the proof of Theorem 1 involves a transcendental inequality of the form $Reg(\hat{\pi}) \leq c_1 Reg(\hat{\pi})^{\frac{\alpha}{2(1+\alpha)}}\tilde{r} + c_2$ for certain positive terms $c_1, c_2$. This inequality is difficult to solve exactly, so we can only get an upper bound on its solution. It turns out that we can get a better upper bound when $2(1 + \alpha)/\alpha$ is an integer.*

**The nuisance estimation rate.** We now show that $Rate^{\textbf{N}}(n, \delta)$ can be effectively controlled for the DM, ISW and DR score functions. In pariticular, $Rate^{\textbf{N}}(n, \delta)$ is can be bounded by the estimation errors of the nuisance functions $f_0$ and $\Sigma_0$.

**Proposition 2.** *For any given $\delta \in (0, 1)$, let $\chi_{n,\delta}$ be a positive sequence converging to 0 as $n \to \infty$, such that the mean square errors of the nuisance estimates satisfy the following with probability at least $1 - \delta$:*

$$\max\left\{\mathbb{E}_X[\|\text{Proj}_{\text{span}(\mathcal{Z})}(\hat{f}(X) - f_0(X))\|^2], \mathbb{E}_X[\|\hat{\Sigma}^\dagger(X) - \Sigma_0^\dagger(X)\|^2_{Fro}]\right\} \leq \chi^2_{n,\delta},$$

*where $\text{Proj}_{\text{span}(\mathcal{Z})}(\hat{f}(X) - f_0(X))$ is the projection of $\hat{f}(X) - f_0(X)$ onto $\text{span}(\mathcal{Z})$. Then,*

1. *If we take $\theta = \theta_{DM}$, we have $Rate^{\textbf{N}}_{DM}(n, \delta) = O(\chi_{n,\delta})$;*

2. *If we take $\theta = \theta_{ISW}$, we have $Rate^{\textbf{N}}_{ISW}(n, \delta) = O(\chi_{n,\delta})$;*

3. *If we take $\theta = \theta_{DR}$, we have $Rate^{\textbf{N}}_{DR}(n, \delta) = O(\chi^2_{n,\delta})$.*

Compared to the DM and ISW scores, the impact of the estimation errors of the nuisances in the DR score is of only second order, *i.e.*, $O(\chi^2_{n,\delta})$ instead of $O(\chi_{n,\delta})$. This echos the benefit of DR methods

in causal inference and offline contextual bandit learning [1, 5, 6]. Notably, here we only need to bound the projected error on the nuisance estimator $\hat{f}$, which handles the setting when span$(\mathcal{Z})$ does not cover the whole $\mathbb{R}^d$ space, as is the case with the shortest path problem in Section 5.

**Computing the critical radius.** The critical radius, $\tilde{r}$ characterizes the complexity of the function classes $\mathcal{G}_1$ and $\mathcal{G}_2$ defined in Eq. (8). The next proposition shows that $\tilde{r}$ is of order $\tilde{O}(1/\sqrt{n})$ if the function classes have finite VC-subgraph dimensions. For simplicity, we focus on $\mathcal{G}_1$ only.

**Proposition 3.** *Suppose $\mathcal{G}_1$ has VC-subgraph dimension $\eta$ almost surely. Then for any $\delta \in (0,1)$, there exists a universal positive constant $\tilde{C}$ such that*

$$\tilde{r} = \tilde{C}\sqrt{\tfrac{\eta \log(n+1)+\log(8/\delta)}{n}} \tag{9}$$

*satisfies the inequalities $\mathcal{R}_n(\mathcal{G}_1, \tilde{r}) \leq \tilde{r}^2$, $3n\tilde{r}^2/64 \geq \log\log_2(1/\tilde{r})$, and $2\exp\left(-3n\tilde{r}^2/64\right) \leq \delta/4$.*

### 3.2 Byproduct: Fast Rates in the Full Feedback Setting with Misspecification

Although our main theorem is stated for the bandit feedback setting, our regret analysis techniques can be easily adapted to the full feedback setting. The following theorem states a similar regret upper bound. To our best knowledge, this is the first result for CLO that shows a margin-dependent fast rate with potential policy misspecification in the full feedback setting.

**Theorem 2.** *Suppose $\mathcal{Z}^*(X)$ defined in Assumption 3 is a singleton almost surely. Define*

$$\mathcal{G}^F = \left\{ (x,y) \to \tfrac{y^\top(\pi(x)-\tilde{\pi}^*(x))\rho}{2B} : \pi \in \Pi_{\mathcal{F}}, \rho \in [0,1] \right\}$$

*and $\tilde{r}^F$ be any solution to the inequality $\mathcal{R}_n(\mathcal{G}^F, r) \leq r^2$ satisfying $3n(\tilde{r}^F)^2/64 \geq \log\log_2(1/\tilde{r}^F)$ and $2\exp\left(-3n(\tilde{r}^F)^2/64\right) \leq \delta$. If Assumption 3 further holds, then, with probability at least $1-\delta$,*

$$Reg(\hat{\pi}^F) \leq B\left(12\sqrt{\tilde{C}(\alpha,\gamma)}\tilde{r}^F\right)^{\frac{2\alpha+2}{\alpha+2}} + 24B\left(\sqrt{\tilde{C}(\alpha,\gamma)}\left(\tfrac{Reg(\tilde{\pi}^*)}{B}\right)^{\frac{\alpha}{2(1+\alpha)}}\tilde{r}^F + (\tilde{r}^F)^2\right) + 2Reg(\tilde{\pi}^*).$$

The regret bound is similar to that in Theorem 1 for the bandit setting, except that it does not have the nuisance error term $2\text{Rate}^{\mathbf{N}}(n/2, \delta/4)$. This is because, in the full feedback setting, we observe the entire $Y$ vector, so we do not need to estimate any nuisance functions and can consider the nuisance estimation error term $\text{Rate}^{\mathbf{N}}(n/2, \delta/4)$ as zero. When $\Pi_{\mathcal{F}}$ is a well-specified VC-subgraph class with dimension $\eta$, we have $Reg(\tilde{\pi}^*) = 0$, and $\tilde{r}^{\mathrm{F}} = \tilde{O}(\sqrt{\eta/n})$, so the bound in Theorem 2 reduces to $O((\eta/n)^{(\alpha+1)/(\alpha+2)} + \eta/n)$. This bound interpolates between $O(n^{-1/2})$ and $O(n^{-1})$ according to the margin parameter $\alpha$, recovering the fast rate in the full-feedback setting without misspecification as given in [14]. In contrast, our bound in Theorem 2 additionally quantifies the impact of policy misspecification, and its generalization Theorem 1 further incorporates the impact of nuisance estimation errors in the bandit-feedback setting.

## 4 Computationally Tractable Surrogate Loss

The IERM objective is generally nonconvex in $f \in \mathcal{F}$, making it computationally intractable to optimize. In the full feedback setting, tractable surrogate losses have been proposed [11, 16]. In this section, we briefly explain the SPO+ loss in [11] and how it can be used in the bandit setting.

The full feedback IERM problem in Eq. (3) can be viewed as minimizing the following loss over $\mathcal{F}$:

$$\min_{f \in \mathcal{F}} \tfrac{1}{n}\sum_{i=1}^n l_{\text{IERM}}(f(X_i), Y_i), \quad \text{where } l_{\text{IERM}}(f(x), y) = y^\top \pi_f(x) - \min_{z \in \mathcal{Z}} y^\top z.$$

This IERM loss is equivalent to the "smart predict-then-optimize" (SPO) loss in Definition 1 of [11]. Letting $z^*(y) \in \arg\min_{z \in \mathcal{Z}} y^\top z$ with the same tie-breaking rule as in $\pi_f$, [11] propose the SPO+ surrogate loss:

$$\min_{f \in \mathcal{F}} \tfrac{1}{n}\sum_{i=1}^n l_{\text{SPO+}}(f(X_i), Y_i), \text{ where } l_{\text{SPO+}}(f(x), y) = \max_{z \in \mathcal{Z}} \; (y - 2f(x))^\top z - (y - 2f(x))^\top z^*(y).$$

The SPO+ loss has many desirable properties: given any fixed $y$, it is an upper bound for the IERM loss, it is convex in $f(x)$, and its subgradient at $f(x)$ has a closed form $2(z^*(y) - z^*(2f(x) - y))$.



Figure 1: Stochastic Shortest path problem on a $5 \times 5$ grid with uncertain edge cost $Y_j$ and decision $z_j$ for $j = 1, \ldots, 40$.

| Methods | Training Data $n$ | | |
|---|---|---|---|
| | 400 | 1000 | 1600 |
| ETO | 3.34% | 0.74% | 0.35% |
| SPO+ DM | 2.30% | 0.36% | 0.16% |
| SPO+ DR PI | 2.47% | 0.59% | 0.32% |
| SPO+ DR Lambda | 2.23% | 0.40% | 0.18% |
| SPO+ DR Clip | 2.29% | 0.44% | 0.20% |
| Naive ETO | 15.03% | 12.12% | 3.53% |
| Naive SPO+ DM | 15.05% | 12.85% | 5.08% |
| Naive SPO+ DR | 14.99% | 13.00% | 5.56% |

Table 1: Average relative regret ratio of different methods over 50 replications when both the policy-inducing model and the nuisance model are correctly specified. The logging policy is a random policy.

In the bandit setting, although $Y_i$ is not observed, the score $\theta(X_i, Z_i, C_i; \hat{f}, \hat{\Sigma})$ plays the same role (see Eqs. (3) and (6)). So it is natural to adapt the SPO+ loss to the bandit setting by replacing the unobserved cost vector $Y_i$ by the corresponding score $\theta(X_i, Z_i, C_i; \hat{f}, \hat{\Sigma})$:

$$\hat{f}_{\text{SPO+}} \in \arg\min_{f \in \mathcal{F}} \ \frac{1}{n} \sum_{j=1,2} \sum_{i \in \mathcal{D}_j} l_{\text{SPO+}}\Big(f(X_i), \theta(X_i, Z_i, C_i; \hat{f}_{3-j}, \hat{\Sigma}_{3-j})\Big).$$

Then we use the plug-in policy $\pi_{\hat{f}_{\text{SPO+}}}$ as the decision policy. This is implemented in the experiments in Section 5. We can similarly adapt any surrogate loss for the full-feedback IERM problem to the bandit feedback setting, simply replacing the cost vector $Y_i$'s by the corresponding scores.

## 5 Numerical Experiments

We now test the performance of our proposed methods in a simulated stochastic shortest path problem following [11, 14]. Specifically, we aim to go from the start node $s$ to the end node $t$ on a $5 \times 5$ grid consisting of $d = 40$ edges, where the costs of traveling on the edges are given by the random vector $Y \in \mathbb{R}^{40}$ (see Fig. 1). We consider covariates $X \in \mathbb{R}^3$ and a function $f_0(x) = \mathbb{E}[Y \mid X = x]$ whose components are cubic polynomials. The corresponding shortest path problem can be easily formulated into a CLO problem with the constraint set $\mathcal{Z}$ given by standard flow preservation constraints. The resulting optimal solution $z$ belongs to $\{0, 1\}^{40}$, indicating whether passing each edge or not. We note that there are $m = 70$ feasible paths $z_1, \ldots, z_m \in \mathcal{Z}$ from the source node to the target node, and the feasible paths are linearly dependent with a rank of 18.

We consider a bandit feedback setting, observing only the total traveling costs $C$ of the historical decisions $Z$ generated by a certain logging policy but not the edge-wise costs $Y$. We consider different types of logging policies that generate the decisions in the observed data. In this section, we report results using a random logging policy that picks a path from all feasible ones at random regardless of the covariate value. In Appendix C, we further study the performance of two different covariate-dependent logging policies: one that picks paths according to the sign of the first covariate $X_1$, and one that depends on the signs of both $X_1$ and $X_2$. The empirical insights from the covariate-dependent logging policies are qualitatively the same as those for the random logging policy.

We numerically evaluate the performance of the ETO approach and the IERM approach[2]. For both approaches, we use the same class $\mathcal{F}$ to construct the decision policies. We consider three different classes for $\mathcal{F}$: a correctly specified polynomial class, a misspecified class that omits two high-order terms (termed degree-2 misspecification), and a misspecified class that omits four high-order terms (termed degree-4 misspecification). For the IERM approach, we consider DM and DR here but defer the results of ISW to Appendix C.2 Table 3 for its significantly worse performance. In both DM and DR, the nuisance $f_0$ is estimated by a bandit-feedback regression given in Eq. (7) with a ridge penalty, and we test out the three aforementioned function classes for the nuisance class $\mathcal{F}^N$ as well. In DR, the nuisance $\Sigma_0(x)$ is estimated by the propensity score approach described in Remark 2,

---

[2]The code scripts for all experiments can be found at `https://github.com/CausalML/CLOBandit`.

with the propensity scores estimated by either sample frequencies (for the random logging policy) or suitable decision tree models (for covariate-dependent logging policies). We further consider three variants when plugging $\hat{\Sigma}$ into the doubly robust score: pseudo-inverse (DR PI) as in the original $\theta_{\text{DR}}$ definition; lambda regularization (DR Lambda), where we replace $\Sigma^{\dagger}$ with $(\Sigma + \lambda I)^{-1}$ for a positive constant $\lambda$; and clipping (DR Clip), where eigenvalues of $\Sigma$ below a certain threshold are clipped to the threshold before taking the pseudo-inverse. For all IERM variants, we optimize the SPO+ losses as discussed in Section 4. Further details on the experimentation setup and implementation are summarized in Appendix C. Finally, we consider some naive extensions of the offline contextual bandit learning with finite discrete actions (Naive ETO and Naive SPO+), where we view the feasible paths as separate discrete actions, without considering the linear structure of the decision-making problem. See Appendix A for details and Section 1.3 for background on offline bandits.

We test the performance of different methods on an independent testing sample of size 2000, and evaluate the ratio of their regrets relative to the expected cost of the global optimal policy $\pi_{f_0}$. Table 1 shows the average relative regret ratios of different methods across 50 replications of the experiment for a random logging policy. Due to space limitations, we include results for the training data size $n = 400, 1000, 1600$, and defer results for other sizes to Appendix C.2. The relative regret of all methods properly decrease with the training data size $n$. In particular, the SPO+ approximation for the end-to-end IERM approach perform better than the ETO method. Among the SPO+ methods, the DM score achieves the best performance, while the DR score based pesudo-inverse performs the worst. Through a close inspection, we found that the bias adjustment term that involves the pseudo-inverse in the DR score causes a significant variance inflation. In fact, the ISW score also performs badly due to the high variance (see Appendix C.2). The Lambda regularization and Clip techniques can effectively reduce the variance and result in improved decision-making. Moreover, the naive benchmarks that ignore the linear structure of the decision costs have much worse performance. This shows the importance of leveraging the linear problem structure.

| | Methods | Training Data $n$ | | | Training Data $n$ | | |
|---|---|---|---|---|---|---|---|
| | | 400 | 1000 | 1600 | 400 | 1000 | 1600 |
| | ETO | $\mathcal{F}$ misspecified degree 2 | | | $\mathcal{F}$ misspecified degree 4 | | |
| | | 11.04% | 9.14% | 8.34% | 12.35% | 11.42% | 10.39% |
| | | $\mathcal{F}$ misspecified degree 2 | | | $\mathcal{F}$ misspecified degree 4 | | |
| Well-specified Nuisance Model $\mathcal{F}^{\text{N}}$ | SPO+ DM | 2.81% | 0.80% | 0.54% | 4.06% | 2.21% | 2.06% |
| | SPO+ DR PI | 3.27% | 1.36% | 1.05% | 4.83% | 2.95% | 2.71% |
| | SPO+ DR Lambda | 2.83% | 0.97% | 0.73% | 4.33% | 2.45% | 2.25% |
| | SPO+ DR Clip | 3.05% | 1.09% | 0.84% | 4.59% | 2.62% | 2.38% |
| | | $\mathcal{F}^{\text{N}}$ misspecified degree 2 | | | $\mathcal{F}^{\text{N}}$ misspecified degree 4 | | |
| Well-specified Policy-inducing Model $\mathcal{F}$ | SPO+ DM | 10.01% | 8.37% | 7.47% | 12.51% | 11.22% | 9.68% |
| | SPO+ DR PI | 9.11% | 7.02% | 6.44% | 11.69% | 10.19% | 9.02% |
| | SPO+ DR Lambda | 9.05% | 7.52% | 6.68% | 12.31% | 10.38% | 8.96% |
| | SPO+ DR Clip | 9.02% | 7.28% | 6.36% | 11.87% | 10.04% | 8.70% |
| | | $\mathcal{F}, \mathcal{F}^{\text{N}}$ misspecified degree 2 | | | $\mathcal{F}, \mathcal{F}^{\text{N}}$ misspecified degree 4 | | |
| Both $\mathcal{F}, \mathcal{F}^{\text{N}}$ Misspecified | SPO+ DM | 9.90% | 8.34% | 7.41% | 12.45% | 11.16% | 9.69% |
| | SPO+ DR PI | 9.15% | 7.23% | 6.52% | 11.92% | 10.46% | 9.42% |
| | SPO+ DR Lambda | 9.03% | 7.46% | 6.74% | 12.01% | 10.72% | 9.25% |
| | SPO+ DR Clip | 8.97% | 7.22% | 6.46% | 11.75% | 10.31% | 8.95% |

Table 2: Mean relative regret ratio of different methods when the nuisance model $\mathcal{F}^{\text{N}}$ and the policy-inducing model $\mathcal{F}$ are misspecified to differrent degrees. The logging policy is a random policy.

In Table 2, we show the performance of different methods when either the policy-inducing model $\mathcal{F}$ or the nuisance model $\mathcal{F}^{\text{N}}$ or both are misspecified. We observe that when the policy-inducing model is misspecified, the end-to-end SPO+ methods perform much better than the ETO method, provided that the nuisance model for SPO+ is correctly specified. This is consistent with the findings in the previous full-feedback CLO literature [e.g., 11, 12, 14], showing the benefit of integrating

estimation and optimization for misspecified policy models. However, the advantage of the end-to-end approaches is weakened dramatically once the nuisance model is misspecified. In this case, the evaluation of decision policies is biased, so the end-to-end approaches also target a "wrong" objective that may not accurately capture the decision quality. Moreover, we observe that when the nuisance model is misspecified, the DR score can somewhat outperform the DM score, because it can debias the misspecified nuisance to some extent. These results demonstrate new challenges with the bandit-feedback CLO: the end-to-end approaches are sensitive to the misspecification of nuisance models, and the DM and DR scores face different bias-and-variance tradeoffs under nuisance model misspecification. Therefore, in practice we may prefer more flexible models for accurate nuisance modeling, while using simpler policy-inducing models for tractable end-to-end optimization.

## 6  Conclusions

This paper studies the bandit-feedback setting for contextual linear optimization for the first time. We adapt the induced empirical risk minimization approach to this setting, provide a novel theoretical analysis for the regret of the resulting policies, leverage surrogate losses for efficient optimization, and empirically demonstrate the performance of the proposed methods across different model specifications. Our paper has a few limitations that we aim to address in the future. First, we primarily consider parametric induced policy classes, and it would be interesting to accommodate more general nonparametric classes. Second, we focus mainly on the SPO+ surrogate loss, and investigating other surrogate losses in the bandit feedback setting would also be of great interest.

## Acknowledgments and Disclosure of Funding

Nathan Kallus acknowledges that this material is based upon work supported by the National Science Foundation under Grant No. 1846210. Xiaojie Mao is supported in part by National Natural Science Foundation of China (grant numbers 72322001, 72201150, and 72293561) and National Key R&D Program of China (grant number 2022ZD0116700).

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

# A   Naive Extensions of Offline Contextual Bandit Learning

In this section, we show some alternative approaches to solve the stochastic shortest path problem in Section 5. These approaches can be viewed as naive extensions of offline contextual bandit learning with discrete actions.

Specifically, consider the feasible paths $z_1, \ldots, z_m$ for $m = 70$. We can view them as separate discrete actions, and a feasible decision policy $\pi$ is a mapping from the covariates $X$ to one of the $m = 70$ feasible paths. We now adopt one-hot-encoding for the decisions. Consider a simplex $\mathcal{Z}^{\text{simplex}}$ in $\mathbb{R}^m$ and zero-one vector $\tilde{z} \in \{0,1\}^m$ that takes the value 1 on one and only one of its coordinate. Each feasible decision in $z_1, \ldots, z_m$ corresponds to one vector $\tilde{z}$, e.g., the decision $z_j$ corresponds to the $\tilde{z}$ vector whose $j$-th entry is 1 and other entries are all 0.

Our decision-making problem restricted to the feasible decisions $z_1, \ldots, z_m$ can be written as follows:

$$\min_{\tilde{z} \in \mathcal{Z}^{\text{simplex}}} \sum_{j=1}^{m} \tilde{z}_j \mathbb{E}[C \mid Z = z_j, X = x] \iff \min_{\tilde{z} \in \mathcal{Z}^{\text{simplex}}} \tilde{z}^\top \tilde{f}_0(x),$$

where $\tilde{f}_0(x) = (\mathbb{E}[C \mid Z = z_1, X = x], \ldots, \mathbb{E}[C \mid Z = z_m, X = x])^\top$. For any given $x$, the resulting decision will be an one-hot vector that corresponds to an optimal decision at $x$ (which can be equivalently given by $\pi_{f_0}(x)$ for an plug-in policy in Eq. (2) at the true $f_0(x) = \mathbb{E}[Y \mid X = x]$). In this formulation, we view the decisions $z_1, \ldots, z_m$ as separate discrete actions, and we do not necessarily take into account the linear structure of the decision cost.

We can easily adapt the bandit-feedback ETO to this new formulation. Specifically, we can first construct an estimator $\hat{\tilde{f}}$ for the $\tilde{f}_0$ function. For offline bandit learning with discrte actions, this would usually be implemented by regressing the observed total cost $C$ with respect to the covariates $X$, within each subsample for each of the feasible decisions respectively. Given the estimator $\hat{\tilde{f}}$, we can then solve the optimization problem $\min_{\tilde{z} \in \mathcal{Z}^{\text{simplex}}} \tilde{z}^\top \hat{\tilde{f}}(x)$. We finally inspect which coordinate of the resulting solution is equal to 1 and choose it as the decision.

To adapt the IERM approach, we similarly define the plug-in policy for any given hypothesis $\tilde{f}(x) : \mathbb{R}^p \to \mathbb{R}^m$ for the function $\tilde{f}_0(x) = (\mathbb{E}[C \mid Z = z_1, X = x], \ldots, \mathbb{E}[C \mid Z = z_m, X = x])^\top$:

$$\tilde{\pi}_{\tilde{f}}(x) \in \arg \min_{\tilde{z} \in \mathcal{Z}^{\text{simplex}}} \tilde{f}(x)^\top \tilde{z},$$

where ties are again broken by some fixed rules. Given a function class $\tilde{\mathcal{F}}$, we can then consider the induced policy class $\tilde{\Pi}_{\tilde{\mathcal{F}}} = \{\tilde{\pi}_{\tilde{f}} : \tilde{f} \in \tilde{\mathcal{F}}\}$. For any policy $\tilde{\pi} \in \tilde{\Pi}_{\tilde{\mathcal{F}}}$, its output is an one-hot vector, whose entry with value 1 corresponds to the chosen decision among $z_1, \ldots, z_m$. For any observed decision $Z \in \{z_1, \ldots, z_m\}$, we denote its one-hot transformation $\tilde{Z}$ as the zero-one vector whose value-one entry corresponds to the value of $Z$. For a given observed total cost $C$, we denote $\tilde{C}$ as the vector all of whose entries are equal to $C$. In the lemma below, we show that the value of each policy $\tilde{\pi}$ can be also identified by some score funtions.

**Lemma 1.** *For any given policy $\tilde{\pi}$ that maps any covariate value $x$ to an $m$-dimensional one-hot vector, its policy value can be written as follows:*

$$V(\tilde{\pi}) = \mathbb{E}\left[ \tilde{\theta}(X, \tilde{Z}, \tilde{C}; \tilde{f}_0, \tilde{e}_0)^\top \tilde{\pi}(X) \right],$$

*where* $\tilde{f}_0(x) = (\mathbb{E}[C \mid Z = z_1, X = x], \ldots, \mathbb{E}[C \mid Z = z_m, X = x])^\top$

$$\tilde{e}_0(x) = (e_0(z_1 \mid x), \ldots, e_0(z_m \mid x)))^\top, \ e_0(z \mid x) = \mathbb{P}(Z = z_j \mid X = x),$$

*and the score function $\tilde{\theta}$ can take three different forms:*

1. *(Direct Method)* $\tilde{\theta}_{DM}(x, \tilde{z}, \tilde{c}; \tilde{f}, \tilde{e}) = \tilde{f}(x)$;

2. *(Inverse Propensity Weighting)* $\tilde{\theta}_{IPW}(x, \tilde{z}, \tilde{c}; \tilde{f}, \tilde{e}) = \frac{\tilde{z}}{\tilde{e}(x)} \tilde{c}$;

3. *(Doubly Robust)* $\theta_{DR}(x, \tilde{z}, \tilde{c}; \tilde{f}, \tilde{e}) = \tilde{f}(x) + \frac{\tilde{z}}{\tilde{e}(x)}(\tilde{c} - \tilde{f}(x))$.

*In the three score functions above, all vector operations are entry-wise operations.*

From Lemma 1, the new formulations above have very similar structure as our previous formulation in Section 2 Proposition 1. The major differences are that we restrict to the discrete actions $z_1, \ldots, z_m$, redefine certain variables accordingly, and consider a simplex set as the constraint. We note that the identification in Lemma 1 mimics the DM, IPW and DR identification in offline contextual bandit learning [e.g., 1, 7, 43]. Since the identification formulae are analogous to those in Section 2, we can easily apply the same policy learning methods in Section 2 and the SPO+ relaxation in Section 4.

# B  Omitted Proofs

## B.1  Supporting Lemmas

**Lemma 2** (Talagrand's inequality, [4, 31, 37]). *Let $U_i, i = 1, \ldots, n$ be independent $\mathcal{U}$-valued random variables. Let $\mathcal{H}$ be a countable family of measurable real-valued functions on $\mathcal{U}$ such that $\|h\|_\infty \leq v$ and $\mathbb{E}[h(U_1)] = \cdots = \mathbb{E}[h(U_n)] = 0$, for all $h \in \mathcal{H}$. Define*

$$v_n = 2v\mathbb{E}\left[\sup_{h \in \mathcal{H}}\left|\sum_{i=1}^n h(U_i)\right|\right] + \sum_{i=1}^n \sup_{h \in \mathcal{H}} \mathbb{E}\left[h^2(U_i)\right].$$

*Then, for all $t \geq 0$,*

$$\mathbb{P}\left(\sup_{h \in \mathcal{H}}\left|\sum_{i=1}^n h(U_i)\right| \geq \mathbb{E}\left[\sup_{h \in \mathcal{H}}\left|\sum_{i=1}^n h(U_i)\right|\right] + t\right) \leq \exp\left(\frac{-t^2}{2v_n + 2tv/3}\right).$$

**Lemma 3.** *Fix functions $f, \Sigma$ independent of $\{(X_i, Z_i, C_i)\}_{i=1}^n$ such that $\|\theta(x, z, c; f, \Sigma)\| \leq \Theta$ for all $x, z, c$. Define the function class*

$$\mathcal{G} = \left\{(x, z, c) \rightarrow \frac{\theta(x, z, c; f, \Sigma)^\top (\pi(x) - \tilde{\pi}^*(x))\rho}{2B\Theta} : \pi \in \Pi_\mathcal{F}, \rho \in [0, 1]\right\}.$$

*Let $\tilde{r}$ be any solution to the inequality $\mathcal{R}_n(\mathcal{G}, r) \leq r^2$ satisfying $3n\tilde{r}^2/64 \geq \log \log_2(1/\tilde{r})$. Then we have*

$$\mathbb{P}\left(\sup_{g \in \mathcal{G}} \frac{|(\mathbb{E}_n - \mathbb{E}_P)g|}{\|g\|_2 + \tilde{r}} \geq 6\tilde{r}\right) \leq 2\exp\left(-\frac{3}{64}n\tilde{r}^2\right),$$

*where*

$$\mathbb{E}_n(g) = \frac{1}{n}\sum_{i=1}^n g(X_i, Z_i, C_i).$$

*Proof of Lemma 3.* When $\sup_{g \in \mathcal{G}}|(\mathbb{E}_n - \mathbb{E}_P)g|/(\|g\|_2 + \tilde{r}) > 6\tilde{r}$, one of the following two events must hold true:

$$\mathcal{E}_1 = \left\{|(\mathbb{E}_n - \mathbb{E}_P)g| \geq 6\tilde{r}^2 \text{ for some } g \in \mathcal{G} \text{ such that } \|g\|_2 \leq \tilde{r}\right\},$$
$$\mathcal{E}_2 = \left\{|(\mathbb{E}_n - \mathbb{E}_P)g| \geq 6\|g\|_2\tilde{r} \text{ for some } g \in \mathcal{G} \text{ such that } \|g\|_2 \geq \tilde{r}\right\}.$$

Define

$$\mathcal{Z}_n(r) = \sup_{g \in \mathcal{G}, \|g\|_2 \leq r} |(\mathbb{E}_n - \mathbb{E}_P)g|.$$

Note that $\|g\|_2 \leq r$ implies $\mathbb{E}[(g - \mathbb{E}_P(g))^2] \leq r^2$, and we also have $\|g - \mathbb{E}_P(g)\|_\infty \leq 2$. By Talagrand's inequality (Lemma 2) over the function class $\{g - \mathbb{E}_P(g) : g \in \mathcal{G}\}$,

$$\mathbb{P}(\mathcal{Z}_n(r) \geq \mathbb{E}[\mathcal{Z}_n(r)] + t) \leq \exp\left(-\frac{nt^2}{8\mathbb{E}[\mathcal{Z}_n(r)] + 2r^2 + 4t/3}\right).$$

We now bound the expectation $\mathbb{E}[\mathcal{Z}_n(r)]$. Since $\mathcal{G}$ is star-shaped[3], by [41, Lemma 13.6], $r \rightarrow \mathcal{R}_n(\mathcal{G}, r)/r$ is non-increasing. Thus, for any $r \geq \tilde{r}$,

$$\mathbb{E}[\mathcal{Z}_n(r)] \leq 2\mathcal{R}_n(\mathcal{G}, r) \leq \frac{2r}{\tilde{r}}\mathcal{R}_n(\mathcal{G}, \tilde{r}) \leq 2r\tilde{r},$$

---

[3]A function class $\mathcal{G}$ is star-shaped if for any $g \in \mathcal{G}$ and $\rho \in [0, 1]$, we have $\rho g \in \mathcal{G}$.

where the first inequality comes from a symmetrization argument, the second inequality uses the fact that $r \to \mathcal{R}_n(\mathcal{G}, r)/r$ is non-increasing, and the third inequality is by definition of $\tilde{r}$.

The Talagrand's then implies

$$\mathbb{P}(\mathcal{Z}_n(r) \geq 2r\tilde{r} + t) \leq \exp\left(-\frac{nt^2}{16r\tilde{r} + 2r^2 + 4t/3}\right). \tag{10}$$

We first bound $\mathbb{P}(\mathcal{E}_1)$. Taking $r = \tilde{r}$ and $t = 4\tilde{r}^2$ in Eq. (10), we get

$$\mathbb{P}(\mathcal{E}_1) \leq \mathbb{P}\big(\mathcal{Z}_n(\tilde{r}) \geq 6\tilde{r}^2\big) \leq \exp\left(-\frac{24}{35}n\tilde{r}^2\right).$$

We now bound $\mathbb{P}(\mathcal{E}_2)$. Note that

$$\mathbb{P}(\mathcal{E}_2) \leq \mathbb{P}(\mathcal{Z}_n(\|g\|_2) \geq 6\|g\|_2\tilde{r} \text{ for some } g \in \mathcal{G}, \|g\|_2 \geq \tilde{r}).$$

Define

$$\mathcal{G}_m = \big\{g \in \mathcal{G} : 2^{m-1}\tilde{r} \leq \|g\|_2 \leq 2^m\tilde{r}\big\}.$$

Since $\|g\|_2 \leq 1$, there exists $M \leq \log_2(1/\tilde{r})$ such that

$$\mathcal{G} \cap \{g : \|g\|_2 \geq \tilde{r}\} \subseteq \cup_{m=1}^M \mathcal{G}_m.$$

Therefore,

$$\mathbb{P}(\mathcal{E}_2) \leq \sum_{m=1}^M \mathbb{P}(\mathcal{Z}_n(\|g\|_2) \geq 6\|g\|_2\tilde{r} \text{ for some } g \in \mathcal{G}_m).$$

We now bound each term in the summation above. If there exists $g \in \mathcal{G}_m$ such that $\mathcal{Z}_n(\|g\|_2) \geq 6\|g\|_2\tilde{r}$, then we have

$$\mathcal{Z}_n(2^m\tilde{r}) \geq \mathcal{Z}_n(\|g\|_2) \geq 6\|g\|_2\tilde{r} \geq 3 \cdot 2^m\tilde{r}^2.$$

Thus,

$$\mathbb{P}(\mathcal{Z}_n(\|g\|_2) \geq 6\|g\|_2\tilde{r} \text{ for some } g \in \mathcal{G}_m) \leq \mathbb{P}\big(\mathcal{Z}_n(2^m\tilde{r}) \geq 3 \cdot 2^m\tilde{r}^2\big).$$

Now, taking $r = 2^m\tilde{r}$ and $t = 2^m\tilde{r}^2$ in Eq. (10), we get

$$\mathbb{P}\big(\mathcal{Z}_n(2^m\tilde{r}) \geq 3 \cdot 2^m\tilde{r}^2\big) \leq \exp\left(-\frac{3n\tilde{r}^2}{13 \cdot 2^{2-m} + 6}\right) \leq \exp\left(-\frac{3}{32}n\tilde{r}^2\right).$$

Therefore, if $\frac{3}{64}n\tilde{r}^2 \geq \log\log_2(1/\tilde{r})$, then

$$\mathbb{P}(\mathcal{E}_2) \leq M\exp\left(-\frac{3}{32}n\tilde{r}^2\right) \leq \exp\left(-\frac{3}{64}n\tilde{r}^2\right).$$

Combining the bounds on $\mathbb{P}(\mathcal{E}_1)$ and $\mathbb{P}(\mathcal{E}_2)$ leads to the final conclusion. $\qquad\square$

**Lemma 4.** *Suppose Assumption 3 holds and $\mathbb{P}(|\mathcal{Z}^*(X)| > 1) = 0$. Then there exists a constant $\tilde{C}(\alpha, \gamma)$ such that for any $\pi \in \Pi_{\mathcal{F}}$,*

$$\mathbb{P}(\pi(X) \neq \pi_{f_0}(X)) \leq \tilde{C}(\alpha, \gamma)\left(\frac{Reg(\pi)}{B}\right)^{\frac{\alpha}{1+\alpha}}.$$

*Proof of Lemma 4.* This follows directly from [14, Lemma 1]. $\qquad\square$

**Lemma 5.** *Let $c_1, c_2, r$ be positive constants. For any $\alpha > 0$, if a positive number $x$ satisfies*

$$x \leq c_1 x^{\frac{\alpha}{2(1+\alpha)}} r + c_2,$$

*we have*

$$x \leq (c_1 r)^{\frac{2\alpha+2}{\alpha+2}} + 2c_2.$$

*Proof of Lemma 5.* First, note that

$$\frac{\partial}{\partial y}\left(y - c_1 y^{\frac{\alpha}{2(1+\alpha)}} r - c_2\right) = 1 - \frac{c_1 r \alpha}{2(1+\alpha)} y^{-\frac{2+\alpha}{2+2\alpha}}.$$

The derivative is strictly increasing in $y$ and is eventually positive. Note the function $y - c_1 y^{\alpha/(2+2\alpha)} r - c_2$ takes a negative value at $y = 0$. Then as $y$ increases, the value of $y - c_1 y^{\alpha/(2+2\alpha)} r - c_2$ first decreases and then increases. Therefore, if $y > 0$ satisfies the inequality $y - c_1 y^{\alpha/(2+2\alpha)} r - c_2 \geq 0$, then such $y$ also provides an upper bound on $x$. Hence it is sufficient to show that $y = (c_1 r)^{\frac{2\alpha+2}{\alpha+2}} + 2c_2$ satisfies the inequality, or equivalently,

$$(c_1 r)^{\frac{2\alpha+2}{\alpha+2}} + c_2 \geq c_1\left((c_1 r)^{\frac{2\alpha+2}{\alpha+2}} + 2c_2\right)^{\frac{\alpha}{2(1+\alpha)}} r. \tag{11}$$

Suppose $\alpha/(2+2\alpha)$ is a rational number. In this case, we can write $\alpha/(2+2\alpha) = m_1/m_2$, where $m_1$ and $m_2$ are positive integers such that $m_2 \geq 2m_1 + 1$. Eq. (11) is then equivalent to

$$\left((c_1 r)^{\frac{m_2}{m_2-m_1}} + c_2\right)^{m_2} \geq c_1^{m_2}\left((c_1 r)^{\frac{m_2}{m_2-m_1}} + 2c_2\right)^{m_1} r^{m_2}. \tag{12}$$

Using the multinomial theorem, we have

$$\left((c_1 r)^{\frac{m_2}{m_2-m_1}} + c_2\right)^{m_2} - c_1^{m_2}\left((c_1 r)^{\frac{m_2}{m_2-m_1}} + 2c_2\right)^{m_1} r^{m_2}$$

$$> \sum_{i=0}^{m_1}\left(\frac{m_2!}{(i+m_2-m_1)!(m_1-i)!} - \frac{m_1! 2^{m_1-i}}{(m_1-i)! i!}\right)(c_1 r)^{\frac{i m_2}{m_2-m_1}+m_2} c_2^{m_1-i}.$$

Since $m_2 \geq 2m_1 + 1$, we have for any $i = 0, \ldots, m_1$,

$$\frac{m_2!}{(i+m_2-m_1)!(m_1-i)!} \geq \frac{m_1! 2^{m_1-i}}{(m_1-i)! i!}.$$

This can be proved by showing the ratio of LHS over RHS is larger than 1. Hence, Eq. (11) holds true when $\alpha/(2+2\alpha)$ is a rational number.

Finally, note that

$$(c_1 r)^{\frac{2\alpha+2}{\alpha+2}} + c_2 - c_1\left((c_1 r)^{\frac{2\alpha+2}{\alpha+2}} + 2c_2\right)^{\frac{\alpha}{2(1+\alpha)}} r$$

is continuous in $\alpha$. Since any real number $\alpha$ can be viewed as the limit of a sequence of rational numbers and Eq. (11) holds for all rational numbers, it also holds true for all $\alpha > 0$. $\qquad\square$

**Lemma 6.** *Let $c_1, c_2, r, y, z$ be positive constants, and let $\alpha$ be a positive constant such that $2(1+\alpha)/\alpha$ is an integer. If a positive number $x$ satisfies*

$$x \leq c_1 x^{\frac{\alpha}{2(1+\alpha)}} r + c_1 y^{\frac{\alpha}{2(1+\alpha)}} r + c_2 r^2 + z + y,$$

*we have*

$$x \leq c_1^{\frac{2\alpha+2}{\alpha+2}} r^{\frac{2\alpha+2}{\alpha+2}} + \frac{2\alpha+2}{\alpha+2} c_1 y^{\frac{\alpha}{2\alpha+2}} r + \frac{2\alpha+2}{\alpha+2} c_2 r^2 + \frac{2\alpha+2}{\alpha+2} z + y.$$

*Proof of Lemma 6.* Let $2(\alpha+1)/\alpha = m$, where $m$ is an integer by assumption. Using similar arguments as in the proof of Lemma 5, it is sufficient to show that for $w = c_1^{\frac{2\alpha+2}{\alpha+2}} r^{\frac{2\alpha+2}{\alpha+2}} + \frac{2\alpha+2}{\alpha+2} c_1 y^{\frac{\alpha}{2\alpha+2}} r + \frac{2\alpha+2}{\alpha+2} c_2 r^2 + \frac{2\alpha+2}{\alpha+2} z + y$ and $c_2' = c_1 y^{\frac{\alpha}{2(1+\alpha)}} r + c_2 r^2 + z + y$, we have

$$w - c_1 w^{\frac{\alpha}{2(1+\alpha)}} r - c_2' \geq 0.$$

This is equivalent to showing

$$c_1^{\frac{2\alpha+2}{\alpha+2}} r^{\frac{2\alpha+2}{\alpha+2}} + \frac{2\alpha+2}{\alpha+2} c_1 y^{\frac{\alpha}{2\alpha+2}} r + \frac{2\alpha+2}{\alpha+2} c_2 r^2 + \frac{2\alpha+2}{\alpha+2} z + y$$

$$\geq c_1\left(c_1^{\frac{2\alpha+2}{\alpha+2}} r^{\frac{2\alpha+2}{\alpha+2}} + \frac{2\alpha+2}{\alpha+2} c_1 y^{\frac{\alpha}{2\alpha+2}} r + \frac{2\alpha+2}{\alpha+2} c_2 r^2 + \frac{2\alpha+2}{\alpha+2} z + y\right)^{\frac{\alpha}{2(1+\alpha)}} r + c_1 y^{\frac{\alpha}{2(1+\alpha)}} r + c_2 r^2 + z + y.$$

Since $\alpha = 2/(m-2)$, the above inequality is equivalent to

$$
\left( c_1^{\frac{m}{m-1}} r^{\frac{m}{m-1}} + \frac{1}{m-1} c_1 y^{\frac{1}{m}} r + \frac{1}{m-1} c_2 r^2 + \frac{1}{m-1} z \right)^m
$$
$$
\geq c_1^m r^m \left( c_1^{\frac{m}{m-1}} r^{\frac{m}{m-1}} + \frac{m}{m-1} c_1 y^{\frac{1}{m}} r + \frac{m}{m-1} c_2 r^2 + \frac{m}{m-1} z + y \right).
$$

Using the multinomial theorem, it is easy to see that the expansion of LHS contains all terms on the RHS (plus additional positive terms). This finishes proving our conclusion. $\square$

## B.2   Proof of Main Theorem

*Proof of Theorem 1.* To simplify notation, we define

$$
\mathbb{E}_{n_j} \left[ \theta(X, Z, C; \hat{f}_{3-j}, \hat{\Sigma}_{3-j})^\top (\hat{\pi}(X) - \tilde{\pi}^*(X)) \right]
$$
$$
= \frac{2}{n} \sum_{i \in \mathcal{D}_j} \theta(X_i, Z_i, C_i; \hat{f}_{3-j}, \hat{\Sigma}_{3-j})^\top (\hat{\pi}(X_i) - \tilde{\pi}^*(X_i)).
$$

We can decompose the regret as

$$
\begin{aligned}
\mathrm{Reg}(\hat{\pi}) =& \mathbb{E}_P \left[ f_0(X)^\top (\hat{\pi}(X) - \pi_{f_0}(X)) \right] \\
=& \mathbb{E}_P \left[ \theta(X, Z, C; f_0, \Sigma_0)^\top (\hat{\pi}(X) - \tilde{\pi}^*(X)) \right] + \mathbb{E} \left[ f_0(X)^\top (\tilde{\pi}^*(X) - \pi_{f_0}(X)) \right] \\
\leq& \frac{1}{2} \sum_{j=1}^2 \mathbb{E}_P \left[ \left( \theta(X, Z, C; f_0, \Sigma_0) - \theta(X, Z, C; \hat{f}_j, \hat{\Sigma}_j) \right)^\top (\hat{\pi}(X) - \tilde{\pi}^*(X)) \right] \\
& + \frac{1}{2} \sum_{j=1}^2 (\mathbb{E}_P - \mathbb{E}_{n_j}) \left[ \theta(X, Z, C; \hat{f}_{3-j}, \hat{\Sigma}_{3-j})^\top (\hat{\pi}(X) - \tilde{\pi}^*(X)) \right] \\
& + \mathrm{Reg}(\tilde{\pi}^*),
\end{aligned}
$$

where the inequality follows from the definition of $\hat{\pi}$.

By Assumption 2, with probability at least $1 - \delta/2$,

$$
\frac{1}{2} \sum_{j=1}^2 \mathbb{E}_P \left[ \left( \theta(X, Z, C; f_0, \Sigma_0) - \theta(X, Z, C; \hat{f}_j, \hat{\Sigma}_j) \right)^\top (\hat{\pi}(X) - \tilde{\pi}^*(X)) \right] \leq \mathrm{Rate}^{\mathbf{N}}(n/2, \delta/4).
$$

We now bound $(\mathbb{E}_P - \mathbb{E}_{n_1}) \left[ \theta(X, Z, C; \hat{f}_2, \hat{\Sigma}_2)^\top (\hat{\pi}(X) - \tilde{\pi}^*(X)) \right]$. By Lemma 3 and the assumption that $2 \exp(-3n\tilde{r}^2/128) \leq \delta/4$, we have that with probability at least $1 - \delta/4$,

$$
\sup_{g \in \mathcal{G}_2} \frac{|(\mathbb{E}_{n_1} - \mathbb{E}_P)g|}{\|g\|_2 + \tilde{r}} \leq 6\tilde{r}. \tag{13}
$$

Assuming Eq. (13) holds,

$$
\begin{aligned}
&(\mathbb{E}_P - \mathbb{E}_{n_1}) \left[ \theta(X, Z, C; \hat{f}_2, \hat{\Sigma}_2)^\top (\hat{\pi}(X) - \tilde{\pi}^*(X)) \right] \\
\leq& 12 B\Theta \left( \left\| \frac{\theta(X, Z, C; \hat{f}_2, \hat{\Sigma}_2)^\top (\hat{\pi}(X) - \tilde{\pi}^*(X))}{2B\Theta} \right\|_2 \tilde{r} + \tilde{r}^2 \right) \\
\leq& 12 B\Theta \left( \left\| \frac{\theta(X, Z, C; \hat{f}_2, \hat{\Sigma}_2)^\top (\hat{\pi}(X) - \pi_{f_0}(X))}{2B\Theta} \right\|_2 \tilde{r} + \left\| \frac{\theta(X, Z, C; \hat{f}_2, \hat{\Sigma}_2)^\top (\pi_{f_0}(X) - \tilde{\pi}^*(X))}{2B\Theta} \right\|_2 \tilde{r} + \tilde{r}^2 \right).
\end{aligned}
$$
$$\tag{14}$$

For any $\pi \in \Pi_{\mathcal{F}}$,

$$\left\| \frac{\theta(X, Z, C; \hat{f}_2, \hat{\Sigma}_2)^\top (\pi(X) - \pi_{f_0}(X))}{2B\Theta} \right\|_2^2$$

$$= \mathbb{E}\left[ \left( \frac{\theta(X, Z, C; \hat{f}_2, \hat{\Sigma}_2)^\top (\pi(X) - \pi_{f_0}(X))}{2B\Theta} \right)^2 \mathbb{I}\{\pi(X) \neq \pi_{f_0}(X)\} \right]$$

$$\leq \mathbb{P}(\pi(X) \neq \pi_{f_0}(X))$$

$$\leq \tilde{C}(\alpha, \gamma) \left( \frac{\mathrm{Reg}(\pi)}{B} \right)^{\frac{\alpha}{1+\alpha}},$$

where the last inequality follows from Lemma 4.

Applying the inequality above for both $\hat{\pi}$ and $\pi_{f_0}$ and plug the bounds into Eq. (14), we get

$$(\mathbb{E}_P - \mathbb{E}_{n_1})\Big[ \theta(X, Z, C; \hat{f}_2, \hat{\Sigma}_2)^\top (\hat{\pi}(X) - \tilde{\pi}^*(X)) \Big]$$

$$\leq 12B\Theta \left( \sqrt{\tilde{C}(\alpha, \gamma)} \left( \frac{\mathrm{Reg}(\hat{\pi})}{B} \right)^{\frac{\alpha}{2(1+\alpha)}} \tilde{r} + \sqrt{\tilde{C}(\alpha, \gamma)} \left( \frac{\mathrm{Reg}(\tilde{\pi}^*)}{B} \right)^{\frac{\alpha}{2(1+\alpha)}} \tilde{r} + \tilde{r}^2 \right).$$

We can similarly bound $(\mathbb{E}_P - \mathbb{E}_{n_2})\Big[ \theta(X, Z, C; \hat{f}_1, \hat{\Sigma}_1)^\top (\hat{\pi}(X) - \tilde{\pi}^*(X)) \Big]$.

Combining all pieces together, we get that with probability at least $1 - \delta$,

$$\frac{\mathrm{Reg}(\hat{\pi})}{B} \leq 12\Theta \left( \sqrt{\tilde{C}(\alpha, \gamma)} \left( \frac{\mathrm{Reg}(\hat{\pi})}{B} \right)^{\frac{\alpha}{2(1+\alpha)}} \tilde{r} + \sqrt{\tilde{C}(\alpha, \gamma)} \left( \frac{\mathrm{Reg}(\tilde{\pi}^*)}{B} \right)^{\frac{\alpha}{2(1+\alpha)}} \tilde{r} + \tilde{r}^2 \right)$$

$$+ \frac{\mathrm{Rate}^{\mathbf{N}}(n/2, \delta/4)}{B} + \frac{\mathrm{Reg}(\tilde{\pi}^*)}{B}.$$

Solving the above inequality with respect to $\mathrm{Reg}(\hat{\pi})/B$ using Lemma 5, we have

$$\mathrm{Reg}(\hat{\pi}) \leq B \left( 12\Theta \sqrt{\tilde{C}(\alpha, \gamma)} \tilde{r} \right)^{\frac{2\alpha+2}{\alpha+2}} + 24B\Theta \left( \sqrt{\tilde{C}(\alpha, \gamma)} \left( \frac{\mathrm{Reg}(\tilde{\pi}^*)}{B} \right)^{\frac{\alpha}{2(1+\alpha)}} \tilde{r} + \tilde{r}^2 \right)$$

$$+ 2\mathrm{Rate}^{\mathbf{N}}(n/2, \delta/4) + 2\mathrm{Reg}(\tilde{\pi}^*).$$

$\square$

## B.3 Proofs of Propositions

*Proof of Proposition 1.* For direct method, the conclusion is obvious.

For ISW, we have for any $\Sigma$,

$$\mathbb{E}\Big[ \big( \Sigma^+(X)ZC \big)^\top \pi(X) \Big] - \mathbb{E}\big[ f_0(X)^\top \pi(X) \big] = \mathbb{E}\big[ f_0(X)^\top (I - \Sigma^+(X)\Sigma_0(X))^\top \pi(X) \big].$$

Let $M(x)$ be a matrix whose columns include all basis vectors of the span of $\mathcal{Z}$. Then $\pi(X) \in$ Range$(M(x))$. According to the coverage assumption, the column space of $\Sigma_0(x)$ is identical to the column space of $M(x)$. By the property of pseudo-inverse, the column space of $(I - \Sigma_0^\dagger \Sigma_0)$ is orthogonal to the column space of $M$. Therefore, $(I - \Sigma^+(X)\Sigma_0(X))^\top \pi(X) = 0$ for any $\pi \in \mathcal{Z}$.

For doubly robust score, we have that for any function $f, \Sigma$,

$$\mathbb{E}\big[ \pi(X)^\top \big( f(X) + \Sigma(X)^+ Z(C - Z^\top f(X)) \big) \big] - \mathbb{E}[\pi(X)^\top f_0(X)]$$

$$= \mathbb{E}\big[ \pi(X)^\top \big( (I - \Sigma^+(X)\Sigma_0(X))(f(X) - f_0(X)) \big) \big].$$

Taking either $f = f_0$ or $\Sigma = \Sigma_0$ gives 0. $\square$

*Proof of Proposition 2.* Because $\pi(x) - \tilde{\pi}^* \in \text{span}(\mathcal{Z})$, for the DM score we have

$$\mathbb{E}_P\left[\left(\theta_{DM}(X,Z,C;f_0,\Sigma_0) - \theta_{DM}(X,Z,C;\hat{f},\hat{\Sigma})\right)^\top (\pi(X) - \tilde{\pi}^*(X))\right]$$

$$\leq \mathbb{E}_P\left[\text{Proj}_{\text{span}(\mathcal{Z})}\left(\theta_{DM}(X,Z,C;f_0,\Sigma_0) - \theta_{DM}(X,Z,C;\hat{f},\hat{\Sigma})\right)^\top (\pi(X) - \tilde{\pi}^*(X))\right]$$

$$\leq 2B\left\{\mathbb{E}_X[\|\text{Proj}_{\text{span}(\mathcal{Z})}(\hat{f}(X) - f_0(X))\|^2]\right\}^{1/2} = O(\chi_{n,\delta}).$$

For the ISW score, we have

$$\mathbb{E}_P\left[\left(\theta_{ISW}(X,Z,C;f_0,\Sigma_0) - \theta_{ISW}(X,Z,C;\hat{f},\hat{\Sigma})\right)^\top (\pi(X) - \tilde{\pi}^*(X))\right]$$

$$\leq 2B\left\{\mathbb{E}_X[\|(\hat{\Sigma}^\dagger(X) - \Sigma_0^\dagger(X))\Sigma_0(X)\|_{\text{Fro}}^2]\right\}^{1/2} = O(\chi_{n,\delta}).$$

For the doubly robust score, we can easily get

$$\mathbb{E}_P\left[\left(\theta_{DR}(X,Z,C;f_0,\Sigma_0) - \theta_{DR}(X,Z,C;\hat{f},\hat{\Sigma})\right)^\top (\pi(X) - \tilde{\pi}^*(X))\right]$$

$$= \mathbb{E}_P\left[(\pi(X) - \tilde{\pi}^*(X))^\top \left(I - \hat{\Sigma}^+(X)\Sigma_0(X)\right)(\hat{f}(X) - f_0(X))\right]$$

$$= \mathbb{E}_P\left[(\pi(X) - \tilde{\pi}^*(X))^\top \left(\Sigma_0^+(X) - \hat{\Sigma}^+(X)\right)\Sigma_0(X)(\hat{f}(X) - f_0(X))\right]$$

$$= \mathbb{E}_P\left[(\pi(X) - \tilde{\pi}^*(X))^\top \left(\Sigma_0^+(X) - \hat{\Sigma}^+(X)\right)\Sigma_0(X)\text{Proj}_{\text{Span}(\mathcal{Z})}(\hat{f}(X) - f_0(X))\right]$$

$$\lesssim \left\{\mathbb{E}_X[\|(\hat{\Sigma}^\dagger(X) - \Sigma_0^\dagger(X))\Sigma_0(X)\|_{\text{Fro}}^2]\right\}^{1/2}\left\{\mathbb{E}_X[\|\text{Proj}_{\text{span}(\mathcal{Z})}(\hat{f}(X) - f_0(X))\|^2]\right\}^{1/2} = O(\chi_{n,\delta}^2).$$

Here the second equation holds because $\pi(x) - \tilde{\pi}^*(x)$ belongs to the linear span of $\mathcal{Z}$, but $I - \Sigma_0^+(x)\Sigma_0$ is orthogonal to the linear span of $\mathcal{Z}$, as we already argued in the proof of Proposition 1. The third equation holds because the column space of $\Sigma_0(X)$ is identical to $\text{span}(\mathcal{Z})$ according to the coverage assumption. $\qquad\square$

*Proof of Proposition 3.* Define

$$\Psi(t) = \frac{1}{5}\exp(t^2).$$

Note that whenever $\mathbb{E}\Psi(|W|/w) \leq 1$ for some random variable $W$, we have by Markov's inequality that

$$\mathbb{P}(|W| > t) \leq 5\exp(-t^2/w^2),$$

$$\mathbb{E}|W| = \int_0^\infty \mathbb{P}(|W| > t)dt \leq 5w. \tag{15}$$

Throughout the proof, we condition on the event that $\mathcal{G}_1$ has VC-subgraph dimension $\eta$. We finish the proof in three steps.

**Step I: Critical radius for empirical Rademacher complexity.** Define the localized empirical Rademacher complexity

$$\hat{\mathcal{R}}_n(\mathcal{G}_1, r) = \mathbb{E}_\epsilon\left[\sup_{g\in\mathcal{G}, \|g\|_n \leq r}\left|\frac{1}{n}\sum_{i=1}^n \epsilon_i g(X_i, Z_i, C_i)\right|\right],$$

where $\epsilon_1, \ldots, \epsilon_n$ are i.i.d. Rademacher random variables, and $\|g\|_n = \sqrt{\sum_{i=1}^n g^2(X_i, Z_i, C_i)/n}$. Let $\hat{r}_n^*$ be the smallest positive solution to $\hat{\mathcal{R}}_n(\mathcal{G}_1, r) \leq r^2/32$. In what follows, we show that there exists a universal constant $C$ such that

$$\mathbb{P}\left(\hat{r}_n^* \leq \tilde{C}\sqrt{\frac{\eta\log(n+1)}{n}}\right) = 1. \tag{16}$$

For any $g \in \mathcal{G}_1$, define set
$$\mathbf{G} = \{(g(X_1, Z_1, C_1), \ldots, g(X_n, Z_n, C_n)) : g \in \mathcal{G}_1, \|g\|_n \leq r\}.$$

Let $D(t, \mathbf{G})$ be the $t$-packing number of $\mathbf{G}$ and $N(t, \mathbf{G})$ be the $t$-covering number. Note that $\|\mathbf{g}\| \leq \sqrt{n}r$ for all $\mathbf{g} \in \mathbf{G}$. By [28, Theorem 3.5],
$$\mathbb{E}_\epsilon \Psi \left( \frac{1}{J} \sup_{g \in \mathcal{G}_1, \|g\|_n \leq r} \left| \sum_{i=1}^n \epsilon_i g(X_i, Z_i, C_i) \right| \right) \leq 1,$$

where
$$J = 9 \int_0^{\sqrt{n}r} \sqrt{\log D(t, \mathbf{G})} dt.$$

So by Eq. (15),
$$\hat{\mathcal{R}}_n(\mathcal{G}_1, r) \leq \frac{5}{n} J.$$

Consider the function class
$$\mathcal{G}_1' = \{g : g \in \mathcal{G}_1, \|g\|_n \leq r\}.$$

Note that $\sqrt{n}r$ is the envelope of $\mathcal{G}_1'$ on $(X_1, Z_1, C_1), \ldots, (X_n, Z_n, C_n)$. Applying [40, Theorem 2.6.7] gives
$$D(\sqrt{n}rt, \mathbf{G}) \leq N(\sqrt{n}rt/2, \mathbf{G})$$
$$\leq \tilde{C}(\eta + 1)(16e)^{\eta+1} \left( \frac{4n}{t^2} \right)^\eta$$

for a universal constant $\tilde{C}$. Thus,
$$J = 9\sqrt{n}r \int_0^1 \sqrt{\log D(\sqrt{n}rt, \mathbf{G})} dt$$
$$\leq 9\sqrt{n}r \int_0^1 \sqrt{\log C + \log(\eta + 1) + (\eta + 1)\log(16e) + \eta \log n + \eta \log 4 - 2\eta \log t} \, dt$$
$$\leq 9 \int_0^1 \sqrt{2 \log C + 15 - 3 \log t} \, dt \sqrt{\eta \log(n+1)n}r,$$

where $\int_0^1 \sqrt{2 \log C + 15 - 3 \log t} \, dt < \infty$. We then obtain that for a (different) universal constant $\tilde{C}$,
$$\hat{\mathcal{R}}_n(\mathcal{G}_1, r) \leq \frac{\tilde{C}}{32} \sqrt{\frac{\eta \log(n+1)}{n}} r.$$

Therefore, for any samples $(X_i, Z_i, C_i)_{i=1}^n$, any $\hat{r}_n \geq C\sqrt{\eta \log(n+1)/n}$ is a valid solution to $\hat{\mathcal{R}}_n(\mathcal{G}_1, r) \leq r^2/32$, which implies Eq. (16).

**Step II: Critical radius for Rademacher complexity.** Let $r_n^*$ be the smallest positive solution to the inequality $\mathcal{R}_n(\mathcal{G}_1, r) \leq r^2/32$. We now bound $r_n^*$.

For any $t > 0$, define the random variable
$$W_n(t) = \mathbb{E}_\epsilon \left[ \sup_{g \in \mathcal{G}_1, \|g\|_2 \leq t} \left| \frac{1}{n} \sum_{i=1}^n \epsilon_i g(X_i, Z_i, C_i) \right| \right],$$

so that $\mathcal{R}_n(\mathcal{G}_1, r) = \mathbb{E}_P[W_n(r)]$ by construction. Define the events
$$\mathcal{E}_3(t) = \left\{ |W_n(t) - \mathcal{R}_n(\mathcal{G}_1, t)| \leq \frac{r_n^* t}{112} \right\},$$

$$\mathcal{E}_4 = \left\{ \sup_{g \in \mathcal{G}_1} \frac{\left| \|g\|_n^2 - \|g\|_2^2 \right|}{\|g\|_2^2 + (r_n^*)^2} \leq \frac{1}{2} \right\}.$$

Following the proof of [14, Lemma EC.12],

$$\mathbb{P}\left(\frac{r_n^*}{5} \leq \hat{r}_n^* \leq 3r_n^*\right) \geq \mathbb{P}(2\mathcal{E}_3(r_n^*) \cap \mathcal{E}_3(7r_n^*) \cap \mathcal{E}_4).$$

[14, Lemma EC.10] implies that

$$\mathbb{P}(\mathcal{E}_4^c) \leq 2e^{-\tilde{c}_1 n(r_n^*)^2}$$

for some universal constant $\tilde{c}_1 > 0$. Moreover, for any $\zeta \geq 1$, we have $\mathcal{R}_n(\mathcal{G}_1, \zeta r_n^*) \geq \mathcal{R}_n(r_n^*) \geq (r_n^*)^2/32$. By [3, Theorem 16],

$$\mathbb{P}(\mathcal{E}_3^c(\zeta r_n^*)) \leq 2e^{-\tilde{c}_2 n(r_n^*)^2}$$

for some universal constant $\tilde{c}_2 > 0$. Combining all pieces we have

$$\mathbb{P}\left(\frac{r_n^*}{5} \leq \hat{r}_n^* \leq 3r_n^*\right) \geq 1 - 6e^{-(\tilde{c}_1 \wedge \tilde{c}_2)n(r_n^*)^2}. \tag{17}$$

By step I in the proof, $\mathbb{P}\left(\hat{r}_n^* \leq \tilde{C}_0\sqrt{\eta \log(n+1)/n}\right) = 1$ for some constant $\tilde{C}_0$. Let $\tilde{C} > 5\tilde{C}_0$ be a constant such that $2^{-\tilde{C}(\tilde{c}_1 \wedge \tilde{c}_2)} < 1/6$. If $r_n^* > C\sqrt{\eta \log(n+1)/n}$, by Eq. (17) we have $\mathbb{P}\left(\hat{r}_n^* > \tilde{C}_0\sqrt{\eta \log(n+1)/n}\right) > 0$, which leads to contradiction. Thus,

$$r_n^* \leq \tilde{C}\sqrt{\eta \log(n+1)/n}.$$

Finally, any $r \geq \tilde{C}\sqrt{\eta \log(n+1)/n}$ solves the inequality $\mathcal{R}_n(\mathcal{G}_1, r) \leq r^2$.

**Step III: Checking other conditions.** The other two inequalities, $3n\tilde{r}^2/64 \geq \log\log_2(1/\tilde{r})$ and $2\exp\left(-3n\tilde{r}^2/64\right) \leq \delta/2$, are easily satisfied as long as we take $\tilde{C}$ big enough. $\qquad\square$

## B.4 Proof for Full Feedback Setting

*Proof of Theorem 1.* To simplify notation, we write

$$\mathbb{E}_n\left[Y^\top\left(\hat{\pi}^{\mathrm{F}}(X) - \tilde{\pi}^*(X)\right)\right] = \frac{1}{n}\sum_{i=1}^{n} Y_i^\top\left(\hat{\pi}^{\mathrm{F}}(X_i) - \tilde{\pi}^*(X_i)\right).$$

We can decompose the regret as

$$\begin{aligned}
\mathrm{Reg}\left(\hat{\pi}^{\mathrm{F}}\right) &= \mathbb{E}_P\left[Y^\top\left(\hat{\pi}^{\mathrm{F}}(X) - \pi_{f_0}(X)\right)\right] \\
&= \mathbb{E}_P\left[Y^\top\left(\hat{\pi}^{\mathrm{F}}(X) - \tilde{\pi}^*(X)\right)\right] + \mathbb{E}\left[Y^\top\left(\tilde{\pi}^*(X) - \pi_{f_0}(X)\right)\right] \\
&\leq (\mathbb{E}_P - \mathbb{E}_n)\left[Y^\top\left(\hat{\pi}^{\mathrm{F}}(X) - \tilde{\pi}^*(X)\right)\right] + \mathrm{Reg}(\tilde{\pi}^*),
\end{aligned}$$

where the inequality follows from the definition of $\hat{\pi}^{\mathrm{F}}$.

We now bound $(\mathbb{E}_P - \mathbb{E}_n)\left[Y^\top\left(\hat{\pi}^{\mathrm{F}}(X) - \tilde{\pi}^*(X)\right)\right]$. Following a similar proof as in the proof of Lemma 3, we have that with probability at least $1 - \delta$,

$$\sup_{g \in \mathcal{G}} \frac{|(\mathbb{E}_n - \mathbb{E}_P)g|}{\|g\|_2 + \tilde{r}^{\mathrm{F}}} \leq 6\tilde{r}^{\mathrm{F}}. \tag{18}$$

Assuming Eq. (18) holds,

$$\begin{aligned}
&(\mathbb{E}_P - \mathbb{E}_n)\left[Y^\top\left(\hat{\pi}^{\mathrm{F}}(X) - \tilde{\pi}^*(X)\right)\right] \\
&\leq 12B\left(\left\|\frac{Y^\top\left(\hat{\pi}^{\mathrm{F}}(X) - \tilde{\pi}^*(X)\right)}{2B}\right\|_2 \tilde{r}^{\mathrm{F}} + (\tilde{r}^{\mathrm{F}})^2\right) \\
&\leq 12B\left(\left\|\frac{Y^\top\left(\hat{\pi}^{\mathrm{F}}(X) - \pi_{f_0}(X)\right)}{2B}\right\|_2 \tilde{r}^{\mathrm{F}} + \left\|\frac{Y^\top\left(\pi_{f_0}(X) - \tilde{\pi}^*(X)\right)}{2B}\right\|_2 \tilde{r}^{\mathrm{F}} + (\tilde{r}^{\mathrm{F}})^2\right).
\end{aligned}$$

For any $\pi \in \Pi_{\mathcal{F}}$,

$$\left\| \frac{Y^\top (\pi(X) - \pi_{f_0}(X))}{2B} \right\|_2^2 = \mathbb{E}\left[ \left( \frac{Y^\top (\pi(X) - \pi_{f_0}(X))}{2B} \right)^2 \mathbb{I}\{\pi(X) \neq \pi_{f_0}(X)\} \right]$$
$$\leq \mathbb{P}(\pi(X) \neq \pi_{f_0}(X))$$
$$\leq \tilde{C}(\alpha, \gamma) \left( \frac{\text{Reg}(\pi)}{B} \right)^{\frac{\alpha}{1+\alpha}},$$

where the last inequality follows from Lemma 4. Thus,

$$(\mathbb{E}_P - \mathbb{E}_n)\left[ Y^\top (\hat{\pi}^{\text{F}}(X) - \tilde{\pi}^*(X)) \right]$$
$$\leq 12B \left( \sqrt{\tilde{C}(\alpha, \gamma)} \left( \frac{\text{Reg}(\hat{\pi}^{\text{F}})}{B} \right)^{\frac{\alpha}{2(1+\alpha)}} \tilde{r}^{\text{F}} + \sqrt{\tilde{C}(\alpha, \gamma)} \left( \frac{\text{Reg}(\tilde{\pi}^*)}{B} \right)^{\frac{\alpha}{2(1+\alpha)}} \tilde{r}^{\text{F}} + (\tilde{r}^{\text{F}})^2 \right).$$

Combining all pieces together, we get that with probability at least $1 - \delta$,

$$\frac{\text{Reg}(\hat{\pi}^{\text{F}})}{B} \leq 12 \left( \sqrt{\tilde{C}(\alpha, \gamma)} \left( \frac{\text{Reg}(\hat{\pi}^{\text{F}})}{B} \right)^{\frac{\alpha}{2(1+\alpha)}} \tilde{r}^{\text{F}} + \sqrt{\tilde{C}(\alpha, \gamma)} \left( \frac{\text{Reg}(\tilde{\pi}^*)}{B} \right)^{\frac{\alpha}{2(1+\alpha)}} \tilde{r}^{\text{F}} + (\tilde{r}^{\text{F}})^2 \right)$$
$$+ \frac{\text{Reg}(\tilde{\pi}^*)}{B}.$$

Solving the above inequality with respect to $\text{Reg}(\hat{\pi}^{\text{F}})/B$ using Lemma 5, we have

$$\text{Reg}(\hat{\pi}^{\text{F}}) \leq B \left( 12\sqrt{\tilde{C}(\alpha, \gamma)} \tilde{r}^{\text{F}} \right)^{\frac{2\alpha+2}{\alpha+2}} + 24B \left( \sqrt{\tilde{C}(\alpha, \gamma)} \left( \frac{\text{Reg}(\tilde{\pi}^*)}{B} \right)^{\frac{\alpha}{2(1+\alpha)}} \tilde{r}^{\text{F}} + (\tilde{r}^{\text{F}})^2 \right) + 2\text{Reg}(\tilde{\pi}^*).$$

$\square$

## C    Additional Experimental Details

In Section 5, we provide experimental results for different methods under various model specifications and different logging policies. In this section, we further explain the details of experiment setup, implementation, and provide additional experimental results. All experiments in the paper are implemented on a cloud computing platform with 128 CPUs of model Intel(R) Xeon(R) Platinum 8369B CPU @ 2.70GHz, 250GB RAM and 500GB storage. The experiment for each specification of $\mathcal{F}$ and $\mathcal{F}^{\text{N}}$ and each logging policy takes around 2 days of running time based on parallel computing on 100 CPUs.

### C.1    Experimental Setup and Implementation Details

**Data generating process.**    We first generate i.i.d draws of the covariates $X = (X_1, X_2, X_3)^\top \in \mathbb{R}^3$ from independent standard normal distribution. Then we simulate the full feedback $Y$ according to the equation $Y = f_0(X) + \epsilon$, where $f_0(X) = 3 + W_1^* X_1 + W_2^* X_2 + W_3^* X_3 + W_4^* X_1 X_2 + W_5^* X_2 X_3 + W_6^* X_1 X_3 + W_7^* X_1 X_2 X_3$ for coefficient vectors $W_1^*, \ldots, W_7^* \in \mathbb{R}^d$ and a random noise $\epsilon$ drawn from the Unif$[-0.5, 0.5]$. To fix the coefficient vectors $W_1^*, \ldots, W_7^*$, we draw their entries independently the from the Unif$[0, 1]$ distribution once, and then use the resulting fixed coefficient vectors throughout the experiment. We sample observed decisions $Z$ from the set of all feasible path decisions $\{z_1, \ldots, z_m\}$ for $m = 70$, according to different logging policies that will be described shortly. Then the total cost $C = Y^\top Z$ is recorded in the observed data.

We consider three different logging policies. One is a random logging policy that uniformly samples each decision from the feasible decisions, regardless of the covariate value. The other two are covariate-dependent logging policies. For these two policies, we first remove 20 feasible decisions that correspond to the optimal decisions for some covariate observations in the testing data, and then randomly sample the observed decisions from the rest of 50 feasible decisions according to different rules. This means that many promising decisions are not explored by the logging policies at all. We hope to use these logging policies to demonstrate the value of leveraging the linear structure of the

decision-making problem, since exploiting the linear structure allows to extrapolate the feedback from the logged decisions to decisions never explored by the policies. Specifically, we further divide the remaining 50 feasible decisions into two even groups. Then the two different logging policies sample decisions from the two groups according to different rules depending on the covariate value.

- One covariate-dependent logging policy samples the decisions according to the sign of the first covariate $X_1$. When $X_1 > 0$, the logging policy chooses the first group with probability $2/3$ and the second group with probability $1/3$. When $X_1 \leq 0$, the logging policy chooses the first group with probability $1/3$ and the second group with probability $2/3$. Once deciding the group, the policy then uniformly samples one decision from the chosen group. For concreteness, we will refer to this policy as the $X_1$-policy.

- The other covariate-dependent policy samples the decisions according to the signs of both $X_1$ and $X_2$. When $X_1 > 0$ and $X_2 > 0$, the logging policy chooses the two groups with probabilities $2/3$ and $1/3$ respectively. When $X_1 > 0$ and $X_2 \leq 0$, the logging policy chooses the two groups with probabilities $1/3$ and $2/3$ respectively. When $X_1 \leq 0$ and $X_2 > 0$, the policy chooses the two groups with probabilities $3/4$ and $1/4$ respectively. When $X_1 \leq 0$ and $X_2 \leq 0$, the policy chooses the two groups with probabilities $1/4$ and $3/4$ respectively. Once deciding the group, the policy again uniformly samples one decision from the chosen group. We will refer to this policy as the $X_1X_2$-policy.

**Specification of the policy-inducing model and nuisance model.** For the policy-inducing model and nuisance model, we consider three different classes. One is the correctly specified class $\{(x_1, x_3, x_3) \mapsto W_0 + W_1 x_1 + W_2 x_2 + W_3 x_3 + W_4 x_1 x_2 + W_5 x_2 x_3 + W_6 x_1 x_3 + W_7 x_1 x_2 x_3 : W_0, \ldots, W_7 \in \mathbb{R}\}$. The second model class $\{(x_1, x_3, x_3) \mapsto W_0 + W_1 x_1 + W_2 x_2 + W_3 x_3 + W_4 x_1 x_2 + W_5 x_2 x_3 : W_0, \ldots, W_5 \in \mathbb{R}\}$ omits two interaction terms and is thus misspecified (which we refer to as degree-2 misspecification). The third model class $\{(x_1, x_3, x_3) \mapsto W_0 + W_1 x_1 + W_2 x_2 + W_3 x_3 : W_1, W_2, W_3 \in \mathbb{R}\}$ omits all four interaction terms (which we refer to as degree-4 misspecification).

**Nuisance estimation for the SPO+ methods.** The SPO+ methods involve two different nuisances. One is the function $f_0(x) = \mathbb{E}[Y \mid X = x]$. We estimate this by the least squares regression in Eq. (7), with the $\mathcal{F}^N$ class being one of the three classes described above (i.e., correct specification, degree-2 misspecification, degree-4 misspecification). We incorporate an additional ridge penalty with a coefficient 1. We estimate the the nuisance $\Sigma_0(x)$ using the propensity score approach described in Remark 2, so we need to estimate the propensity scores $\mathbb{P}(Z = z_j \mid X = x)$ for $j = 1, \ldots, m$. For the random logging policy, we simply estimate $\mathbb{P}(Z = z_j \mid X = x)$ for any $x$ by the empirical frequency of the decision $z_j$ in the observed data. For the $X_1$-policy and $X_1X_2$-policy, we estimate the propensity scores by classification decision trees of depth 3 trained to classify each instance to one of the observed classes among $z_1, \ldots, z_m$. These nuisances are all estimated using the two-fold cross-fitting described in Section 1.2.

Since the $\Sigma_0$ matrix is rank-deficient, we cannot directly invert it. Besides taking the pseudo-inverse, we also implement the Lambda regularization and clipping techniques described in Section 5. For Lambda regularization, we set the regularization parameter $\lambda$ to 1. For the clipping technique, we consider the eigen-decomposition of $\Sigma_0$, set all eigenvalues below 1 to 1, and then take a pseudo-inverse of the transformed matrix. These two lead to the SPO+ DR Lambda and SPO+ DR Clip methods in Section 5 and this section.

**Naive Benchmarks.** We also implemented the benchmarks in Appendix A. For ETO, SPO+ DM, and SPO+ DR, we estimate $\tilde{f}_0(x)$ by regressing $C$ with respect to $X$ using data for each observed decision respectively. The regression function class uses similar correctly specified class, degree-2 misspecified class, degree-4 misspecified class mentioned above, with only slight difference in the dimension since the output of $\tilde{f}_0$ is $m$-dimensional while the output of $f_0$ is $d$-dimensional. For SPO+ DR and SPO+ IPW, we need to estimate the propensity scores. These are again estimated by either sample frequency or decision trees. Note that some feasible decisions are never observed in the training data. For these decisions, the corresponding component of $\tilde{f}_0$ is heuristically imputed by a pooled regression of $C$ against $X$ using all observed data. For SPO+ IPW and SPO+ DR, although the propensity scores for the unseen decisions are zero, they do not impact the policy evaluation since they only need the propensity score for decisions observed in the data.

**SPO+ Tuning and Optimization.** We need to solve the SPO+ optimization problem in Section 4 for the DM and DR scores. Following [11], we incorporate an additional ridge penalty on the coefficients of the hypotheses. We select the penalty coefficient from a grid of $0, 0.001, 0.01$ and $10$ points distributed uniformly on the logarithmic scale over $0.1$ to $100$. This is done by minimizing the out-of-sample error on an independent validation dataset with size equal to the corresponding training data. The penalty coefficient is finally set as the half of the value chosen by this validation procedure.

To optimize the SPO+ loss, [11] recommend applying standard convex optimization solvers to a dual reformulation, or by running stochastic subgradient descents. We tried both approaches and found that the stochastic subgradient descent method with the default hyperparameters in [11, 14] (number of iterations, step size, batch size) runs much faster, while performing similarly to the dual reformulation method.

## C.2 Additional Experimental Results

In this section, we provide some additional experimental results. Specifically, Tables 3 and 4 are extensions of Tables 1 and 2 for the random logging policy. In particular, Table 3 shows additional results on ISW and IPW methods. They are generally worse than other alternative methods. Table 4 additionally show results for the naive benchmarks under model misspecification, and further confirm their worse performance. Tables 5 to 8 provide results for two covariate-dependent logging policies, the $X_1$-policy and $X_1 X_2$-policy respectively. The performance of different methods tend to remain similar or slightly degrade under these more complicated logging policies. However, the comparisons of different methods remain qualitatively the same.

| Methods | Training Data $n$ | | | | | | |
|---|---|---|---|---|---|---|---|
| | 400 | 600 | 800 | 1000 | 1200 | 1400 | 1600 |
| ETO | 3.34% | 1.86% | 1.04% | 0.74% | 0.50% | 0.41% | 0.35% |
| SPO+ DM | 2.30% | 1.14% | 0.64% | 0.36% | 0.25% | 0.20% | 0.16% |
| SPO+ DR PI | 2.47% | 1.44% | 0.86% | 0.59% | 0.45% | 0.39% | 0.32% |
| SPO+ DR Lambda | 2.23% | 1.15% | 0.64% | 0.40% | 0.27% | 0.23% | 0.18% |
| SPO+ DR Clip | 2.29% | 1.22% | 0.69% | 0.44% | 0.31% | 0.25% | 0.20% |
| SPO+ ISW | 15.00% | 15.52% | 15.11% | 15.02% | 14.98% | 15.03% | 15.11% |
| Naive ETO | 15.03% | 15.26% | 14.74% | 12.12% | 8.45% | 5.17% | 3.53% |
| Naive SPO+ DM | 15.05% | 15.46% | 14.92% | 12.85% | 9.81% | 6.63% | 5.08% |
| Naive SPO+ DR | 14.99% | 15.42% | 14.97% | 13.00% | 10.10% | 6.98% | 5.56% |
| Naive SPO+ IPW | 15.56% | 15.77% | 16.06% | 16.06% | 16.06% | 15.96% | 15.96% |

Table 3: Mean relative regret ratio of different methods when the nuisance model $\mathcal{F}^{\mathrm{N}}$ and the policy-inducing model $\mathcal{F}$ are correctly specified. The logging policy is a random policy.

| | Methods | Training Data $n$ | | | Training Data $n$ | | |
|---|---|---|---|---|---|---|---|
| | | 400 | 1000 | 1600 | 400 | 1000 | 1600 |
| | ETO | $\mathcal{F}$ misspecified degree 2 | | | $\mathcal{F}$ misspecified degree 4 | | |
| | | 11.04% | 9.14% | 8.34% | 12.35% | 11.42% | 10.39% |
| | | $\mathcal{F}$ misspecified degree 2 | | | $\mathcal{F}$ misspecified degree 4 | | |
| Well-specified Nuisance Model $\mathcal{F}^{\mathrm{N}}$ | SPO+ DM | 2.81% | 0.80% | 0.54% | 4.06% | 2.21% | 2.06% |
| | SPO+ DR PI | 3.27% | 1.36% | 1.05% | 4.83% | 2.95% | 2.71% |
| | SPO+ DR Lambda | 2.83% | 0.97% | 0.73% | 4.33% | 2.45% | 2.25% |
| | SPO+ DR Clip | 3.05% | 1.09% | 0.84% | 4.59% | 2.62% | 2.38% |
| | Naive SPO+ DM | 14.97% | 12.78% | 5.68% | 15.27% | 13.20% | 6.42% |
| | Naive SPO+ DR | 15.00% | 13.03% | 6.31% | 15.27% | 13.48% | 7.26% |
| | | $\mathcal{F}^{\mathrm{N}}$ misspecified degree 2 | | | $\mathcal{F}^{\mathrm{N}}$ misspecified degree 4 | | |
| Well-specified Policy-inducing Model $\mathcal{F}$ | SPO+ DM | 10.01% | 8.37% | 7.47% | 12.51% | 11.22% | 9.68% |
| | SPO+ DR PI | 9.11% | 7.02% | 6.44% | 11.69% | 10.19% | 9.02% |
| | SPO+ DR Lambda | 9.05% | 7.52% | 6.68% | 12.31% | 10.38% | 8.96% |
| | SPO+ DR Clip | 9.02% | 7.28% | 6.36% | 11.87% | 10.04% | 8.70% |
| | Naive SPO+ DM | 15.56% | 14.23% | 12.96% | 15.22% | 14.51% | 13.85% |
| | Naive SPO+ DR | 15.64% | 14.36% | 13.31% | 15.17% | 14.67% | 14.12% |
| | | $\mathcal{F}, \mathcal{F}^{\mathrm{N}}$ misspecified degree 2 | | | $\mathcal{F}, \mathcal{F}^{\mathrm{N}}$ misspecified degree 4 | | |
| Both $\mathcal{F}, \mathcal{F}^{\mathrm{N}}$ Misspecified | SPO+ DM | 9.90% | 8.34% | 7.41% | 12.45% | 11.16% | 9.69% |
| | SPO+ DR PI | 9.15% | 7.23% | 6.52% | 11.92% | 10.46% | 9.42% |
| | SPO+ DR Lambda | 9.03% | 7.46% | 6.74% | 12.01% | 10.72% | 9.25% |
| | SPO+ DR Clip | 8.97% | 7.22% | 6.46% | 11.75% | 10.31% | 8.95% |
| | Naive SPO+ DM | 15.65% | 14.23% | 13.02% | 15.16% | 14.53% | 13.84% |
| | Naive SPO+ DR | 15.63% | 14.42% | 13.33% | 15.26% | 14.74% | 13.95% |

Table 4: Mean relative regret ratio of different methods when the nuisance model $\mathcal{F}^{\mathrm{N}}$ and the policy-inducing model $\mathcal{F}$ are misspecified to differrent degrees. The logging policy is a random policy.

| Methods | Training Data $n$ | | | | | | |
|---|---|---|---|---|---|---|---|
| | 400 | 600 | 800 | 1000 | 1200 | 1400 | 1600 |
| ETO | 3.20% | 1.83% | 1.05% | 0.72% | 0.56% | 0.47% | 0.38% |
| SPO+ DM | 2.16% | 1.12% | 0.60% | 0.40% | 0.30% | 0.25% | 0.20% |
| SPO+ DR PI | 2.51% | 1.50% | 0.97% | 0.73% | 0.59% | 0.50% | 0.44% |
| SPO+ DR Lambda | 2.14% | 1.10% | 0.62% | 0.41% | 0.34% | 0.28% | 0.23% |
| SPO+ DR Clip | 2.15% | 1.18% | 0.65% | 0.44% | 0.37% | 0.31% | 0.26% |
| SPO+ ISW | 15.49% | 15.39% | 15.46% | 15.43% | 15.63% | 15.75% | 15.75% |
| Naive ETO | 15.44% | 14.92% | 10.77% | 6.56% | 4.59% | 3.39% | 2.87% |
| Naive SPO+ DM | 15.46% | 15.21% | 11.56% | 7.49% | 5.57% | 4.32% | 3.67% |
| Naive SPO+ DR | 15.47% | 15.19% | 11.91% | 8.09% | 6.04% | 4.81% | 4.18% |
| Naive SPO+ IPW | 15.57% | 15.67% | 15.71% | 15.74% | 15.66% | 15.79% | 15.80% |

Table 5: Mean relative regret ratio of different methods when the nuisance model $\mathcal{F}^{\mathrm{N}}$ and the policy-inducing model $\mathcal{F}$ are correctly specified. The logging policy is a $X_1$-policy.

| | Methods | Training Data $n$ | | | Training Data $n$ | | |
|---|---|---|---|---|---|---|---|
| | | 400 | 1000 | 1600 | 400 | 1000 | 1600 |
| | ETO | $\mathcal{F}$ misspecified degree 2 | | | $\mathcal{F}$ misspecified degree 4 | | |
| | | 11.55% | 9.56% | 8.68% | 11.41% | 10.03% | 9.60% |
| | | $\mathcal{F}$ misspecified degree 2 | | | $\mathcal{F}$ misspecified degree 4 | | |
| Well-specified Nuisance Model $\mathcal{F}^{\mathrm{N}}$ | SPO+ DM | 2.86% | 1.10% | 0.86% | 4.19% | 2.53% | 2.27% |
| | SPO+ DR PI | 3.57% | 1.74% | 1.39% | 5.11% | 3.36% | 3.10% |
| | SPO+ DR Lambda | 2.93% | 1.26% | 1.03% | 4.45% | 2.79% | 2.53% |
| | SPO+ DR Clip | 3.05% | 1.34% | 1.11% | 4.57% | 2.98% | 2.66% |
| | Naive SPO+ DM | 15.59% | 7.80% | 4.37% | 15.43% | 8.08% | 5.19% |
| | Naive SPO+ DR | 15.60% | 8.45% | 5.00% | 15.38% | 8.83% | 5.90% |
| | | $\mathcal{F}^{\mathrm{N}}$ misspecified degree 2 | | | $\mathcal{F}^{\mathrm{N}}$ misspecified degree 4 | | |
| Well-specified Policy-inducing Model $\mathcal{F}$ | SPO+ DM | 10.47% | 8.31% | 7.68% | 10.77% | 9.37% | 8.82% |
| | SPO+ DR PI | 9.32% | 7.48% | 6.97% | 10.69% | 9.80% | 9.43% |
| | SPO+ DR Lambda | 9.15% | 7.48% | 6.98% | 10.36% | 9.10% | 8.72% |
| | SPO+ DR Clip | 9.23% | 7.37% | 6.78% | 10.37% | 8.94% | 8.46% |
| | Naive SPO+ DM | 15.63% | 14.58% | 13.78% | 14.93% | 13.08% | 12.71% |
| | Naive SPO+ DR | 15.56% | 14.54% | 13.95% | 14.92% | 13.57% | 13.30% |
| | | $\mathcal{F}, \mathcal{F}^{\mathrm{N}}$ misspecified degree 2 | | | $\mathcal{F}, \mathcal{F}^{\mathrm{N}}$ misspecified degree 4 | | |
| Both $\mathcal{F}, \mathcal{F}^{\mathrm{N}}$ Misspecified | SPO+ DM | 10.36% | 8.27% | 7.58% | 10.66% | 9.35% | 8.90% |
| | SPO+ DR PI | 9.19% | 7.21% | 6.46% | 10.35% | 8.96% | 8.55% |
| | SPO+ DR Lambda | 9.24% | 7.47% | 6.98% | 10.36% | 9.18% | 8.75% |
| | SPO+ DR Clip | 9.38% | 7.39% | 6.76% | 10.28% | 8.94% | 8.55% |
| | Naive SPO+ DM | 15.61% | 14.56% | 13.88% | 14.96% | 13.25% | 12.77% |
| | Naive SPO+ DR | 15.49% | 14.62% | 14.12% | 14.96% | 13.48% | 13.23% |

Table 6: Mean relative regret ratio of different methods when the nuisance model $\mathcal{F}^{\mathrm{N}}$ and the policy-inducing model $\mathcal{F}$ are misspecified to differrent degrees. The logging policy is a $X_1$-policy.

| Methods | Training Data $n$ | | | | | | |
|---|---|---|---|---|---|---|---|
| | 400 | 600 | 800 | 1000 | 1200 | 1400 | 1600 |
| ETO | 3.09% | 1.92% | 1.28% | 0.86% | 0.58% | 0.41% | 0.34% |
| SPO+ DM | 2.19% | 1.19% | 0.66% | 0.43% | 0.32% | 0.24% | 0.19% |
| SPO+ DR PI | 2.49% | 1.42% | 1.00% | 0.72% | 0.60% | 0.46% | 0.45% |
| SPO+ DR Lambda | 2.08% | 1.15% | 0.67% | 0.46% | 0.35% | 0.26% | 0.21% |
| SPO+ DR Clip | 2.17% | 1.21% | 0.73% | 0.51% | 0.36% | 0.30% | 0.25% |
| SPO+ ISW | 14.57% | 14.65% | 14.49% | 14.04% | 14.36% | 14.26% | 14.30% |
| Naive ETO | 14.30% | 13.99% | 11.01% | 7.55% | 5.21% | 3.76% | 3.11% |
| Naive SPO+ DM | 14.31% | 14.21% | 11.89% | 8.44% | 6.16% | 4.69% | 3.97% |
| Naive SPO+ DR | 14.29% | 14.23% | 12.12% | 8.77% | 6.52% | 5.12% | 4.45% |
| Naive SPO+ IPW | 15.52% | 15.64% | 15.64% | 15.66% | 15.76% | 15.74% | 15.70% |

Table 7: Mean relative regret ratio of different methods when the nuisance model $\mathcal{F}^{\mathrm{N}}$ and the policy-inducing model $\mathcal{F}$ are correctly specified. The logging policy is a $X_1 X_2$-policy.

| | Methods | Training Data $n$ | | | Training Data $n$ | | |
|---|---|---|---|---|---|---|---|
| | | 400 | 1000 | 1600 | 400 | 1000 | 1600 |
| | ETO | $\mathcal{F}$ misspecified degree 2 | | | $\mathcal{F}$ misspecified degree 4 | | |
| | | 10.08% | 8.86% | 8.18% | 12.03% | 11.15% | 11.03% |
| | | $\mathcal{F}$ misspecified degree 2 | | | $\mathcal{F}$ misspecified degree 4 | | |
| Well-specified Nuisance Model $\mathcal{F}^{\mathrm{N}}$ | SPO+ DM | 2.91% | 1.10% | 0.83% | 4.32% | 2.56% | 2.24% |
| | SPO+ DR PI | 3.55% | 1.73% | 1.42% | 5.27% | 3.36% | 3.04% |
| | SPO+ DR Lambda | 2.91% | 1.28% | 1.01% | 4.50% | 2.78% | 2.49% |
| | SPO+ DR Clip | 3.07% | 1.37% | 1.12% | 4.67% | 2.94% | 2.69% |
| | Naive SPO+ DM | 14.32% | 8.92% | 4.69% | 14.52% | 9.32% | 5.46% |
| | Naive SPO+ DR | 14.34% | 9.33% | 5.37% | 14.56% | 9.78% | 6.13% |
| | | $\mathcal{F}^{\mathrm{N}}$ misspecified degree 2 | | | $\mathcal{F}^{\mathrm{N}}$ misspecified degree 4 | | |
| Well-specified Policy-inducing Model $\mathcal{F}$ | SPO+ DM | 8.78% | 7.90% | 7.11% | 11.93% | 11.08% | 10.55% |
| | SPO+ DR PI | 8.23% | 6.97% | 6.51% | 11.91% | 11.36% | 11.07% |
| | SPO+ DR Lambda | 8.28% | 7.14% | 6.60% | 11.92% | 11.01% | 10.38% |
| | SPO+ DR Clip | 8.19% | 7.04% | 6.37% | 11.72% | 10.95% | 10.36% |
| | Naive SPO+ DM | 14.86% | 12.50% | 12.07% | 14.68% | 13.84% | 13.09% |
| | Naive SPO+ DR | 14.77% | 12.78% | 12.26% | 14.65% | 14.19% | 13.70% |
| | | $\mathcal{F}, \mathcal{F}^{\mathrm{N}}$ misspecified degree 2 | | | $\mathcal{F}, \mathcal{F}^{\mathrm{N}}$ misspecified degree 4 | | |
| Both $\mathcal{F}, \mathcal{F}^{\mathrm{N}}$ Misspecified | SPO+ DM | 8.75% | 7.83% | 7.09% | 11.95% | 11.07% | 10.53% |
| | SPO+ DR PI | 8.07% | 6.66% | 6.03% | 11.79% | 10.84% | 10.40% |
| | SPO+ DR Lambda | 8.23% | 7.14% | 6.58% | 11.90% | 10.87% | 10.43% |
| | SPO+ DR Clip | 8.09% | 6.95% | 6.41% | 11.90% | 10.74% | 10.22% |
| | Naive SPO+ DM | 14.85% | 12.66% | 12.14% | 14.70% | 14.02% | 13.11% |
| | Naive SPO+ DR | 14.75% | 12.78% | 12.26% | 14.67% | 14.17% | 13.41% |

Table 8: Mean relative regret ratio of different methods when the nuisance model $\mathcal{F}^{\mathrm{N}}$ and the policy-inducing model $\mathcal{F}$ are misspecified to differrent degrees. The logging policy is a $X_1 X_2$-policy.

