# OpenReview forum: "Contextual Linear Optimization with Bandit Feedback"
_NeurIPS.cc/2024/Conference — NeurIPS 2024 poster_

### Official Review · Reviewer_h4hQ · 2024-07-12

**Soundness:** 3
**Presentation:** 3
**Contribution:** 3
**Rating:** 6
**Confidence:** 3

**Summary:**

This paper studies Contextual Linear Optimization (CLO) with bandit feedback and presents a class of algorithms termed Induced Empirical Risk Minimization (IERM). The authors also derive the regret upper bounds. The regret analysis accounts for the misspecification of the policy class and incorporates a margin condition that potentially enables a faster regret rate.

**Strengths:**

1. The extension of the CLO problem to bandit feedback is an important and novel research direction.
2. The mathematical formulations and proofs are rigorous and well-presented.
3. The analysis allows for model misspecification in the induced policy class.

**Weaknesses:**

Although I believe this paper is technically sound, the readability could be improved. The presentation of the problem and model could be clearer, especially for readers unfamiliar with the CLO problem. It would be better to provide a more detailed explanation of the problem setting before delving into mathematical formulas. For example, in the introduction, the authors could explain the interaction protocol between the learner and the environment, and specify which quantities are known or unknown to the learner/environment. Such elaborations would make the paper more accessible.

**Questions:**

N/A

**Limitations:**

Limitations are adequately discussed in the paper.

---

> ### Author Rebuttal · Authors · 2024-08-07
>
> Thank you for your comments. We agree that providing more background information on the CLO problem would be helpful. Here, we clarify the unknown quantities and the interaction between the learner and the environment.
>
> In classic CLO problems, the learner aims to solve the optimization problem $\min_{z\in\mathcal{Z}}\mathbb{E}[Y^\top z \mid X = x] = f_0(x)^\top z$ for all observed context values $x$, where $f_0(x)$ is the conditional expectation function $\mathbb{E}[Y \mid X = x]$. This means the learner seeks to minimize the expected  total decision cost $Y^\top z$ given the observed context value $x$.
> The true  distribution of $(Y, X)$ is unknown, making the unknown $f_0$ function the main challenge for the learner.
>
>
> Before making decisions, the learner  needs to estimate the $f_0$ function from samples $(X_i, Y_i)_{i = 1}^n$, typically assumed to be i.i.d draws from the population distribution of $(X, Y)$. Note that this is an offline setting: the learner is provided with a batch of existing data $(X_i, Y_i)$ for $i=1, ..., n$ and uses this dataset for learning, without further interaction with the environment to generate more data while learning and making decisions.  The existing literature has considered two major types of approaches to learning $f_0$ and the corresponding decision policy:  estimate-then-optimize (ETO) approaches and integrated approaches. The latter directly targets the decision objective and is thus favored by many recent papers.
>
> Our paper studies a CLO problem where the cost  coefficient $Y$ is not directly observed in the data. Instead, we observe only the total costs of some historical decisions implemented in the past. Specifically, we have access to some historical decisions $Z_1, \dots, Z_n$, made by some previous policies that might depend on the corresponding contexts $X_1, \dots, X_n$ and potentially other unobserved factors. The total costs $C_i = Y_i^\top Z_i$ for these decisions were recorded,  but the underlying cost coefficient $Y_1, \dots, Y_n$ were not. This means the learner only observes the costs of the decisions that were actually implemented, without information on the costs of other unimplemented counterfactual decisions. These constitute the observed samples $(X_i, Z_i, C_i)_{i = 1}^n$. Again, we consider an offline learning setting where the learner is given this dataset collected under  previous policies, without further interacting with the environment to generate new data. The learner does not know the population distribution of $(X, Z, Y, C)$ and aims to use the given dataset  $(X_i, Z_i, C_i)$ for $i=1, ..., n$ to learn a new decision policy for future decision-making.
> This is akin to the offline contextual bandits widely studied in the literature [1, 2, 3]. However, the offline contextual bandit literature mostly considers the optimization over finitely many arms, while we consider a more general constrained linear programming problem. We in particular study how to extend the integrated IERM approach from regular CLO problems to CLO with partial/bandit feedback.
>
> In summary, the main unknown quantity to the learner in the CLO problems is the conditional expectation function $f_0(x) = \mathbb{E}[Y \mid X = x]$ of the cost coefficient $Y$ given contextual features $X$. Once this function is known,  the CLO problems can be solved easily. Classic CLO literature considers learning this unknown function from samples of the cost coefficients and the contexts $(X_i, Y_i)$. Our paper considers the setting where only total costs $C_i$ associated with historical decisions $Z_i$ (along with the contexts $X_i$) can be observed, but the underlying cost coefficients $Y_i$'s are unobserved.
> Both these CLO literature and our paper consider an offline setting where the learner is given a fixed set of data to decide a policy, without further interacting with the environment to collect more data.
> It would be interesting to consider online settings where the learner can directly interact with the environment, generating data on the fly while learning and making decisions. But the online learning is out of the scope of our current paper.
>
>
> Thanks for your suggestions. We will incorporate these clarifications into our Section 1. In particular, we will highlight that our paper considers an offline learning setting in both the abstract and introduction.
>
>
> [1] Dudík, Miroslav, John Langford, and Lihong Li. "Doubly robust policy evaluation and learning." Proceedings of the 28th International Conference on International Conference on Machine Learning. 2011.
>
> [2] Dudík, Miroslav, et al. "Doubly Robust Policy Evaluation and Optimization." Statistical Science 29.4 (2014): 485-511.
>
> [3] Swaminathan, Adith, and Thorsten Joachims. "Batch learning from logged bandit feedback through counterfactual risk minimization." The Journal of Machine Learning Research 16.1 (2015): 1731-1755.

---

> > ### Comment · Reviewer_h4hQ · 2024-08-10
> >
> > Thanks for your responses and my concerns have been addressed.

---

> > > ### Author Response · Authors · 2024-08-11
> > >
> > > Dear Reviewer h4hQ. Thank you for reading our rebuttal. As you write that it has addressed all your concerns, we would greatly appreciate if you would raise your rating and confidence scores accordingly. Thank you. And do let us know if you have further questions -- we would do our best to answer them promptly.

---

> > > > ### Comment · Reviewer_h4hQ · 2024-08-12
> > > >
> > > > Thank you for your response. I appreciate the contributions your paper makes and I am willing to support this paper. However, to maintain consistency and fairness, I have decided to keep my current score. This score is based on a balanced comparison with other papers in my review batch.

---

### Official Review · Reviewer_d35g · 2024-07-12

**Soundness:** 4
**Presentation:** 3
**Contribution:** 4
**Rating:** 8
**Confidence:** 4

**Summary:**

This work addresses the partial bandit feedback setup for contextual linear optimization. The authors propose an algorithm based on Induced Empirical Risk Minimization (IERM), which incorporates doubly-robust estimation to handle the reward model misspecification. They provide a regret bound for the partial bandit feedback setup and, as a byproduct, offer a regret bound for the full bandit feedback setup under model misspecification. This contribution is significant due to the robustness of the theoretical results and the broad applicability of the findings. The paper is well-written and merits acceptance with only minor revisions.

**Strengths:**

This work accommodates the practicality such as partial feedback and model misspecification. In addition, the theoretical results are solid enough. The numerical experiments demonstrate the theoretical results well.

**Weaknesses:**

This setup is similar to partial monitoring. It would be great if the suggested model is compared with partial monitoring. In addition, it is hard to interpret the result to the reviewer, as the complexities of terms are not stated. The reviewer would like to see a summary of what the order of regret with respect to n is. Further, the model misspecification is not fully explained in the first two sections. The authors should explain the model misspecification before starting a theoretical analysis. Lastly, there are some minor errors in the presentation:
- In line 108, "is" should be corrected.
- There is no definition of $\chi^2_{n,\delta}$.
- In line 74, "in that that" should be corrected.

**Questions:**

Based on my understanding, ETO is a family of algorithms that use estimates directly, such as plug-in policies with estimates. In contrast, IERM-type algorithms include an additional optimization step based on these estimates. This step appears to make IERM-type algorithms function similarly to optimism in the face of uncertainty (OFU)-type algorithms. What aspects of IERM-type algorithms make them more effective than ETO-type algorithms? Is this improvement related to their exploration schemes?

**Limitations:**

It is not well explained why the suggested algorithm has improved performance as compared to the existing algorithms.

---

> ### Author Rebuttal · Authors · 2024-08-07
>
> Thanks for your comments. We now address each point individually.
>
> **Comparison to partial monitoring.** It is interesting to compare our problem with partial monitoring. There are at least two major differences. One is that partial monitoring usually considers an online setting where the learner interacts with the environment and collects data on the fly while making decisions. In contrast, we consider an offline setting where the learner is provided with a fixed dataset without further interaction. See our response to Reviewer h4hQ for more details. The other is that partial monitoring is designed to allow the observed feedback to be different from the reward associated with each decision (a.k.a, arm) [1]. However, our setting is closer to the typical bandit setting, since our  observed feedback $C = Y^\top Z$ is exactly the decision cost (i.e., negative reward) of each decision $Z$.
>
> **Complexity terms in theoretical guarantees.** Thank you for raising this issue. Our regret bounds are formulated in terms of the critical radius, a widely used complexity measure in statistical learning theory. This measure is very general and recovers the information-theoretically optimal rate for many function classes [2]. However, we agree that this may be abstract and could impede the readers' understanding. As we explain below Theorem 1, a concrete example is when the function classes $\mathcal{G}_1, \mathcal{G}_2$ are VC subgraph classes with VC dimension $\eta$. The corresponding critical radius is $\tilde{r} = O(\sqrt{\eta/n})$. Then the regret bound in Theorem 1 reduces to
> \begin{align*}
>    \text{Reg}(\hat{\pi}) \lesssim \left(\frac{\eta}{n}\right)^{\frac{\alpha + 1}{\alpha + 2}} + \text{Reg}(\tilde{\pi}^*) \left(\frac{\eta}{n}\right)^{\frac{1}{2}} + \frac{\eta}{n} + \text{Rate}^N(n/2, \delta/4) +  \text{Reg}(\tilde{\pi}^*).
> \end{align*}
> The regret rate depends on the degree of misspecification characterized by $\text{Reg}(\tilde{\pi}^*)$ (i.e, the regret of the best-in-policy-class policy), the nuisance estimation error rate $\text{Rate}^N(n/2, \delta/4)$, and the margin parameter $\alpha$. Hu et al. [3] study the full-feedback setting without misspecification, so there is no nuisance estimation error or misspecification error (i.e., $\text{Rate}^N(n/2, \delta/4) = 0$ and $\text{Reg}(\tilde{\pi}^*) = 0$). Our bound recovers their fast rate  $O((\eta/n)^{(\alpha+1)/(\alpha+2)} + \eta/n)$ that interpolates between $O(n^{-1/2})$ and $O(n^{-1})$ according to the margin parameter $\alpha$. Our new bound shows that this fast rate could be dominated by the slow rate $O(\text{Reg}(\tilde{\pi}^*) ({\eta}/{n})^{{1}/{2}})$ when there is significant misspecification. Moreover, our bound shows that in the bandit-feedback setting, we also need to estimate nuisance functions, and the corresponding error $\text{Rate}^N(n/2, \delta/4)$ can also affect the final regret bound. The order of this nuisance error rate depends on the form of the score function (see Proposition 2) and the complexity of estimating the nuisances. For example, if we use the doubly robust score and both nuisances can be estimated at rates $ o(n^{-1/4})$, then $\text{Rate}^N(n/2, \delta/4) = o(n^{-1/2})$ is negligible [4]. We plan to explain these clearly in a dedicated remark  in Section 3.
>
> **Model misspecification.** Thanks for your suggestion of explaining the model misspecification before our theoretical analysis. We totally agree with this suggestion, because accommodating model misspecification is one of our major technical contributions. We plan to  explain model misspecification in Section 1.1 when we first introduce the induced policy class.
>
> **What makes IERM-type algorithms more effective?** IERM can be more effective than ETO because it more directly targets the decision-making problem. ETO first constructs an estimator $\hat f$ for the unknown function $f_0(x) = \mathbb{E}[Y \mid X = x]$ by fitting standard regressions (e.g., linear regressions), and then solve optimization $\min_{z\in\mathcal{Z}}\hat f(x)^\top z$ for each context value $x$ to get the final   policy. However, the estimation step typically optimizes some regression error (e.g., least squares objective), without considering the downstream decision-making problem.
>
> In contrast, IERM considers the policies $\pi_f \in \text{argmin}_{z\in\mathcal{Z}}f(x)^\top z$ induced by candidate regression functions $f$ (e.g., linear functions) and then selects the policy  achieving the smallest in-sample average  cost. IERM directly optimizes the decision cost objective, thereby achieving better decision performance in many scenarios. This is quite different from the optimism in face of uncertainty type algorithms for balancing the exploration and exploitation in online decision-making.
>
> **Minor errors.** Thanks for pointing out the typos, which we will correct in the future. Regarding the definition of $\chi^2_{n, \delta}$, it is a vanishing sequence used to characterize the error in estimating the two nuisance functions. We will change the first two lines of Proposition 2 to the following: ''For any given $\delta \in (0, 1)$, let $\chi_{n, \delta}$ be a positive sequence converging to $0$ as $n \to \infty$, such that the mean square errors of the nuisance estimates satisfy the following with probability at least $1-\delta$''.
>
> [1] Kirschner, Johannes, Tor Lattimore, and Andreas Krause. "Linear partial monitoring for sequential decision making: Algorithms, regret bounds and applications." The Journal of Machine Learning Research (2023).
>
> [2] Wainwright, Martin J. High-dimensional statistics: A non-asymptotic viewpoint. Cambridge university press, 2019.
>
> [3] Hu, Yichun, Nathan Kallus, and Xiaojie Mao. "Fast rates for contextual linear optimization." Management Science (2022).
>
> [4] Victor Chernozhukov, Mert Demirer, Greg Lewis, and Vasilis Syrgkanis. Semi-parametric efficient policy learning with continuous actions. NeurIPS 2019.

---

> > ### Comment · Reviewer_d35g · 2024-08-14
> >
> > I thank the authors for their detailed response.

---

### Official Review · Reviewer_wdeG · 2024-07-14

**Soundness:** 3
**Presentation:** 2
**Contribution:** 3
**Rating:** 6
**Confidence:** 1

**Summary:**

The paper aims at the following problem. The learner observes side information $x$ and needs to output $z$. The Nature will then generate $Y$ (which depends on $x$, but may be randomised) and the learner will suffer loss $Y'z$. The key feature of the setup is that $Y$ is not directly observable. The paper describes convincing real-life applications fitting this scenario.

One is after a policy $z=\pi(x)$ minimising the loss expectation on the basis of the past data. One can obtain it by estimating the conditional expectations $f_0(x) = E(Y\mid x)$ first. However, this is not necessarily the best approach. In the style of reinforcement learning, the paper aims to estimate the policy value $V(\pi)$ directly.

The paper puts forward an algorithm, or, rather an approach for this called IERM (I understand some steps in it involve optimisation, which is very hard to perform). The performance of the approach is studied in Theorem 1. It bounds the regret of the policy found by IERM w.r.t. the optimal policy in a class. The bound involves Rademacher complexity of the class.

There is substantial empirical evaluation.

**Strengths:**

I believe the approach of the paper is useful and very practical.

**Weaknesses:**

The paper is very hard to read for an uninitiated person like me. I found it very hard to distinguish between given data, estimates, datapoints, random variables etc.

Perhaps it would help if the paper were structured in a more traditional way with definitions and algorithms rather than the free-flowing narrative.

Page 3, line 125: I am not sure I parse

which we call as nuisance functions as they will be used to

correctly. Did you mean

which we call nuisance functions because they will be used to

?

(I should say I do not understand the choice of the word "nuisance". The pun is lost on me.)

**Questions:**

None.

**Limitations:**

Yes

---

> ### Author Rebuttal · Authors · 2024-08-07
>
> Thank you for your feedback. We now further clarify our problem and explain the data, estimates, random variables, and nuisances involved.
>
> First, let's consider a linear programming problem $\min_{z\in \mathcal{Z}} y^\top z$ where $z$ is the decision variable, $\mathcal{Z}$ is the constraint set, and $y$ is the coefficient vector in the objective. This can model many problems, such as the shortest-path problem in our paper. Specifically, consider finding the shortest path from a starting point to a destination along edges on a graph (with $d$ edges in total). Any feasible path can be represented by a vector $z \in \\{0, 1\\}^d$, with each entry corresponding to one edge and a value of 1 indicating that the edge is picked by the path. A vector $y$ gives the traveling time along each edge respectively. Thus, the total traveling time for a path $z$ is $y^\top z$. The set $\mathcal{Z}$ constrains $z$ to represent a path (or, a convex combination of paths) such that $\text{argmin}_{z\in\mathcal{Z}}y^\top z$ gives a path that achieves the shortest total traveling time.
>
> Our paper considers an uncertain environment where the objective coefficient is a random vector $Y$. For example, the traveling time could be stochastic and uncertain in practice. While the realized value of $Y$ is unknown at the time of decision-making, we may have access to some predictive contextual features $X$ (also considered a random vector). For example, although we do not know the actual traveling time in each shortest-path decision-making instance, we may observe features such as weather, time of day, and traffic conditions that are predictive of the traveling time. Contextual linear optimization (CLO) provides a formal way to incorporate features in decision-making, solving for the optimal decision that minimizes the expected cost upon observing each context value $x$:
> \begin{align*}
> \min_{z\in\mathcal{Z}}~ \mathbb{E}\left[Y \mid X = x\right]^\top z.
> \end{align*}
> Since the distribution of $(X, Y)$ is unknown in practice, existing CLO literature uses samples $(X_i, Y_i)_{i=1}^n$ to estimate the unknown function $f_0(x) = \mathbb{E}[Y \mid X = x]$. There are primarily two different approaches: the estimate-then-optimize approach and the integrated approach. The estimate-then-optimize approach directly fits a standard regression of $Y$ against $X$ to get an estimator $\hat f$, usually by minimizing some statistical error measures like mean squared errors. The integrated approach considers the decision policies $\pi_f$ induced by each candidate estimator $f$ (e.g., linear functions):
>
> \begin{align*}
> \pi_f(x) \in \text{argmin}_{z\in\mathcal{Z}}~ f(x)^\top z.
> \end{align*}
>
> Then it picks an optimal policy achieving the smallest sample average cost $\frac{1}{n}\sum_{i=1}^n Y_i^\top \pi_f(X_i)$.
> The integrated approach directly targets the decision cost objective and is favored in recent literature. Our paper also focuses on the integrated approach.
>
> The existing literature only considers the full-feedback setting where $Y$ is fully observed so its observations can be directly used to evaluate any decision policy. In contrast, we consider a partial or bandit feedback setting where we can only observe the total cost $C = Y^\top Z$ for some historical decision $Z$ prescribed by a previous policy. Here the historical decision $Z$ is also considered as a random vector because it may have been prescribed according to $X$ or some other uncertain factors. Therefore, we have access to samples $(X_i, Z_i, C_i)_{i=1}^n$ but without observing the underlying cost coefficients $Y_i$'s. In this case, we cannot directly evaluate the sample average cost of any induced policy $\pi_f$.
>
> Our Proposition 1 shows that under Assumption 1, we can evaluate the mean cost of a policy $\pi_f$ by using a score function $\theta$ in place of the unobserved $Y$. This score function may depend on not only the observed variables but also two unknown functions $f_0$ and $\Sigma_0$. We thus propose to first construct some estimators $\hat f$ and $\hat \Sigma$ for the unknown functions and then plug these estimators into the score function for the downstream evaluation and learning of decision policies. This process involves constructing estimators $\hat f$ and $\hat \Sigma$ in the score function and the estimator for the policy value $\frac{1}{n}\sum_{i=1}^n \theta(X_i, Z_i, C_i; \hat f, \hat \Sigma)^\top \pi_f(X_i)$. Our Section 2 considers a slightly different procedure that incorporates additional sample splitting, but the overall idea is the same.
>
> We refer to the unknown functions $f_0$ and $\Sigma_0$ in the score function $\theta$ as "nuisance" functions, because these functions are not directly used for decision-making  but merely serve as intermediaries for the evaluation of the policy value. This is standard terminology in statistics and is also frequently used in off-policy evaluation and learning. For example, the off-policy evaluation of a contextual bandit policy or reinforcement learning policy often involves estimating a behavior policy function (propensity score) or a conditional reward Q function. These are often referred to as nuisance functions as well [1].
>
> Thank you for your comments again. We will further clarify the data, estimators and random variables and explain the meaning of ''nuisance'' in the camera-ready version.
>
> [1] Uehara, Masatoshi, Chengchun Shi, and Nathan Kallus. "A review of off-policy evaluation in reinforcement learning." arXiv preprint arXiv:2212.06355 (2022).

---

> > ### Comment · Reviewer_wdeG · 2024-08-13
> >
> > Thank you for the detailed response. It helps my understanding.
> >
> > Special thanks for explaining "nuisance". This needs to be incorporated in the paper as the meaning seems to be remote from that in everyday English.

---

### Author Rebuttal · Authors · 2024-08-07

Dear Reviewers,

Thank you for your thoughtful comments and constructive suggestions. We are encouraged to hear that you find our theoretical results solid, our numerical experiments demonstrate the theoretical results well, and our approach  useful and practical.

We appreciate your suggestions for improving the readability and presentation of our paper. We have addressed each of your points individually, and we will incorporate these clarifications in the camera-ready version if our paper is accepted. We are committed to enhancing the presentation and writing of our paper and are grateful for your valuable feedback.

Please let us know if you have any further comments. We are happy to discuss them.


Best regards,

The Authors.

---

### Decision · Program_Chairs · 2024-09-25

**Decision:**

Accept (poster)

**Comment:**

This paper studies the contextual linear optimization problem under 'bandit feedback', i.e. only one projection of the random cost coefficient vector where the projection depends on which decision was made. Some reviewers had more expertise in the area than others, but everyone was positive about the technical contributions of this work. One criticism that was raised by multiple reviewers was that the paper was challenging to read for someone who is uninitiated in the CLO problem (even if they are familiar with adjacent areas, such as online learning/bandits/statistics/causal inference). Given that NeurIPS sees a broad audience, I strongly encourage the authors to make every effort possible to improve readability for the camera-ready version, starting with the reviewers' constructive suggestions.